# ANT: Adaptive Noise Schedule for Time Series Diffusion Models

**Seunghan Lee, Kibok Lee**,[*] **Taeyoung Park**[*]
Department of Statistics and Data Science, Yonsei University
{seunghan9613,kibok,tpark}@yonsei.ac.kr

## Abstract

Advances in diffusion models for generative artificial intelligence have recently propagated to the time series (TS) domain, demonstrating state-of-the-art performance on various tasks. However, prior works on TS diffusion models often borrow the framework of existing works proposed in other domains without considering the characteristics of TS data, leading to suboptimal performance. In this work, we propose Adaptive Noise schedule for Time series diffusion models (ANT), which *automatically* predetermines proper noise schedules for given TS datasets based on their statistics representing non-stationarity. Our intuition is that an optimal noise schedule should satisfy the following desiderata: 1) It linearly reduces the non-stationarity of TS data so that all diffusion steps are equally meaningful, 2) the data is corrupted to the random noise at the final step, and 3) the number of steps is sufficiently large. The proposed method is practical for use in that it eliminates the necessity of finding the optimal noise schedule with a small additional cost to compute the statistics for given datasets, which can be done offline before training. We validate the effectiveness of our method across various tasks, including TS forecasting, refinement, and generation, on datasets from diverse domains. Code is available at this repository: `https://github.com/seunghan96/ANT`.

## 1 Introduction

Diffusion models have demonstrated outstanding performance in generative tasks across diverse domains, and various methods have been proposed in the time series (TS) domain to address a range of tasks, including forecasting, imputation, and generation [26, 34, 38, 18, 30, 1]. However, recent works primarily focus on determining *which architecture* to use for a diffusion model, overlooking the useful information from the domain knowledge for other components, e.g., *noise schedule*.

Noise schedules control the noise added to the data across the diffusion process, with the choice of schedule being crucial for performance [4]. Several works have explored the design of noise schedules [24, 21]; however, they do not consider the characteristics of data, resulting in suboptimal performance, especially in the TS domain. Figure 1 shows the forecasting performance of various models including our method applied to TSDiff [15] on the M4 dataset [23]. Among the various schedules, our method chooses a cosine schedule for the M4 dataset, yielding a 27.8% gain compared to a linear schedule of TSDiff, which is a common schedule across TS datasets. This highlights the importance of selecting an appropriate schedule for each dataset.

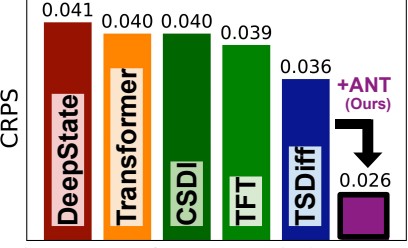

Figure 1: Performance gain by ANT.

In this work, we propose **A**daptive **N**oise schedule for **T**ime series diffusion models (ANT), a method for choosing an adaptive noise schedule based on the statistics representing non-stationarity of the TS dataset. These statistics measure the patterns appeared in the TS, with TS from the real world exhibiting high non-stationarity while white noise TS exhibiting low non-stationarity. We argue that a desirable schedule should gradually transform non-stationary TS into stationary ones, in line with

---

[*]Equal advising.

38th Conference on Neural Information Processing Systems (NeurIPS 2024).

prior research suggesting that the noise level at each step should remain consistent [24]. Furthermore, we opt for a schedule that corrupts the TS into random noise at the final step to align the training and inference stages [21], and a schedule that has a sufficient number of diffusion steps to enhance sample quality [31]. Specifically, we discover that a schedule which decreases the non-stationarity of TS on a *linear scale* corrupts the TS into random noise gradually, making TS across steps more distinguishable, i.e., making each step equally meaningful. Figure 2a shows the forward process using two schedules: the base schedule of TSDiff (w/o ANT) and the schedule proposed by our method (w/ ANT) which aims to decrease the non-stationarity linearly.

Additionally, we investigate the usefulness of diffusion step embedding (DE) in TS diffusion models and argue that it is not necessary when using a linear schedule, as the step information is inherent in the data. We also discover that a non-linear schedule is more robust to total diffusion steps ($T$) compared to a linear schedule, yielding consistent performance across various resource constraints on $T$. The main contributions of this paper are summarized as follows:

- We introduce ANT, an algorithm designed to select an appropriate noise schedule, which is 1) **adaptive** in that the schedule is selected based on the statistics of datasets, 2) **flexible** in that any noise schedule can be a candidate, 3) **model-agnostic** in that it only depends on the statistics of datasets, and 4) **efficient** in that no training is required, as the statistics can be precomputed offline.

- We study the components of diffusion models regarding noise schedules, arguing that diffusion step embedding is unnecessary when using a linear schedule. Additionally, we find that a non-linear schedule exhibits greater robustness to the number of diffusion steps compared to a linear schedule.

- We provide extensive experimental results across various datasets, demonstrating that our method outperforms the baseline in a range of tasks, including TS forecasting, refinement, and generation.

## 2 Background and Related Works

**Denoising diffusion probabilistic model (DDPM).** DDPM [12] is a well-known diffusion model where input $\mathbf{x}^0$ is corrupted to a Gaussian noise during the forward process and $\mathbf{x}^0$ is denoised from $\mathbf{x}^T$ during the backward process with total $T$ diffusion steps. For the forward process, $\mathbf{x}^t$ is corrupted from $\mathbf{x}^{t-1}$ iteratively with Gaussian noise of variance $\beta_t \in [0, 1]$ :

$$q\left(\mathbf{x}^t \mid \mathbf{x}^{t-1}\right) = \mathcal{N}\left(\mathbf{x}^t; \sqrt{1 - \beta_t}\mathbf{x}^{t-1}, \beta_t\mathbf{I}\right), \quad t = 1, \ldots, T. \tag{1}$$

Using a property of Gaussian transition kernel, the forward process of multiple steps can be written as $q\left(\mathbf{x}^t \mid \mathbf{x}^0\right) = \mathcal{N}\left(\mathbf{x}^t; \sqrt{\bar{\alpha}_t}\mathbf{x}^0, (1 - \bar{\alpha}_t)\mathbf{I}\right)$, where $\bar{\alpha}_t = \Pi_{s=1}^t \alpha_s$ and $\alpha_t = 1 - \beta_t$. For the backward process, $\mathbf{x}^{t-1}$ is denoised from $\mathbf{x}^t$ by sampling from the following distribution:

$$p_\theta\left(\mathbf{x}^{t-1} \mid \mathbf{x}^t\right) = \mathcal{N}\left(\mathbf{x}^{t-1}; \mu_\theta\left(\mathbf{x}^t, t\right), \Sigma_\theta\left(\mathbf{x}^t, t\right)\right), \tag{2}$$

where $\mu_\theta\left(\mathbf{x}^t, t\right)$ is defined by a neural network and $\Sigma_\theta\left(\mathbf{x}^t, t\right)$ is usually fixed as $\sigma_t^2\mathbf{I}$. DDPM formulates this task as a noise estimation problem, where $\varepsilon_\theta$ predicts the noise added to $\mathbf{x}^t$. With the predicted noise $\varepsilon_\theta\left(\mathbf{x}^t, t\right)$, $\mu_\theta\left(\mathbf{x}^t, t\right)$ can be obtained by

$$\mu_\theta\left(\mathbf{x}^t, t\right) = \frac{1}{\sqrt{\alpha_t}}\left(\mathbf{x}^t - \frac{1 - \alpha_t}{\sqrt{1 - \bar{\alpha}_t}}\varepsilon_\theta\left(\mathbf{x}^t, t\right)\right), \tag{3}$$

and $\varepsilon_\theta$ is optimized using $\mathcal{L}_\varepsilon = \mathbb{E}_{t,\mathbf{x}^0,\varepsilon}\left[\|\varepsilon - \varepsilon_\theta\left(\mathbf{x}^t, t\right)\|^2\right]$ as a training objective.

**Unconditional TS diffusion models.** Unlike most TS diffusion models which are conditional models [26, 38, 18, 30, 1, 6, 29, 40, 19], unconditional TS diffusion models do not use conditions (i.e. information of the observed values $\mathbf{x}_{\text{obs}}^0$) as explicit inputs during the training stage. Instead, they utilize them as guidance during the inference stage through a self-guidance mechanism. The backward process of unconditional models with the self-guidance term can be expressed as

$$p_\theta\left(\mathbf{x}^{t-1} \mid \mathbf{x}^t, \mathbf{x}_{\text{obs}}^0\right) = \mathcal{N}\left(\mathbf{x}^{t-1}; \mu_\theta\left(\mathbf{x}^t, t\right) + s\sigma_t^2\nabla_{\mathbf{x}^t}\log p_\theta\left(\mathbf{x}_{\text{obs}}^0 \mid \mathbf{x}^t\right), \sigma_t^2\mathbf{I}\right), \tag{4}$$

where $s$ is the scale parameter controlling the self-guidance term. As the self-guidance mechanism helps avoid the need for architectural changes depending on the condition or task, we apply our method to TSDiff [15], which is the SOTA unconditional TS diffusion model. However, our method is not limited to the unconditional model; application to a conditional model, such as CSDI [34], is also discussed in Appendix M.

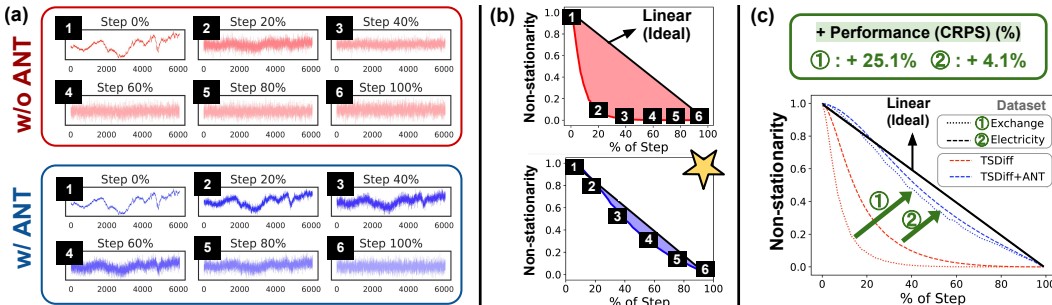

Figure 2: **Overall framework of ANT.** (a) shows that a base schedule (w/o ANT) abruptly corrupts TS at the earlier diffusion step, while a schedule proposed by ANT gradually corrupts it until the final step. (b) visualizes the non-stationarity curves of both schedules and their discrepancy from a linear line, with the schedule that gradually decreases the non-stationarity (w/ ANT) showing lower discrepancy. (c) shows that better performance is achieved as the curves get closer to a linear line.

**Noise schedules for diffusion models.** The choice of the noise schedule is crucial for the performance of diffusion models [4], and various methods have been proposed in the computer vision domain [12, 32, 13], including a cosine schedule [24] to maintain the noise level at each step consistent, and rescaling method [21] to achieve zero signal-to-noise ratio (SNR) at the terminal step to address the misalignment between the training and inference stages. However, these methods are handcrafted for each dataset, without considering the characteristics of TS datasets. Moreover, recent TS diffusion models overlook the importance of noise schedules, treating them merely as hyperparameters, which motivates us to develop a noise schedule specifically tailored for TS datasets using their characteristics.

**Non-stationarity of TS.** The non-stationarity of a TS indicates how different it is from white noise, and various statistics using autocorrelation (AC) exist to assess this. Integrated autocorrelation time (IAT) [22], defined as $\tau_{\text{IAT}} = 1 + 2\sum_{k=1}^{\infty} \rho_k$ where $\rho_k$ is AC at lag $k$, quantifies AC over different lags. Additionally, lag-one autocorrelation (Lag1AC) quantifies AC between adjacent observations, and variance of autocorrelation (VarAC) measures the variance of AC over different lags [37]. These statistics, which capture the noisiness of TS, inform the speed at which the TS collapses into noise during the diffusion process. In this paper, we propose a new statistic based on IAT that captures the non-stationarity of TS, accounting for both positive and negative AC.

## 3 Methodology

In this section, we introduce our main contribution, ANT, an adaptive noise schedule for TS diffusion models. ANT proposes a noise schedule resembling an ideal one that gradually diminishes the non-stationarity of TS as the diffusion step progresses. Subsequently, we investigate the properties of commonly used noise schedules and argue that: 1) The diffusion step embedding (DE) is unnecessary for TS diffusion models when a linear schedule is employed, and 2) non-linear schedules are more robust to the change of the number of diffusion steps than linear schedules in terms of the performance and the scale parameter of the self-guidance mechanism of unconditional diffusion models.

### 3.1 ANT: Adaptive Noise Schedule for TS Diffusion Models

The overall framework of ANT is illustrated in Figure 2, where we aim to find a noise schedule that decreases the non-stationarity of TS on a linear scale. Figure 2a shows how the non-stationarity of a TS gradually decreases with ANT, while it decreases abruptly without ANT. This difference can be explained by computing the non-stationarity curve of a TS for a given noise schedule as shown in Figure 2b. ANT proposes a noise schedule that minimizes the discrepancy between the ideal linear line and the non-stationarity curve of the schedule, whose $x$-axis and $y$-axis represent the progress of the step (%) and the (normalized) statistics of non-stationarity, respectively. As illustrated in Figure 2c, reduction in discrepancy with ANT yields better performance compared to without ANT.

**Statistics of non-stationarity.** While various statistics can be used to quantify the non-stationarity of TS, we propose the *integrated absolute autocorrelation time (IAAT)*. IAAT is a variation of IAT [22] that takes the absolute value of the autocorrelation to account for positive and negative correlations without canceling them out: $\tau_{\text{IAAT}} = 1 + 2\sum_{k=1}^{\infty} |\rho_k|$. Although IAAT shows the best performance, we note that ANT is robust across different statistics as shown in Table 8.

**Adaptive schedule.** Figure 3 illustrates the non-stationarity curves of various schedules for two datasets [17], with line width indicating total diffusion steps. The figure reveals that different

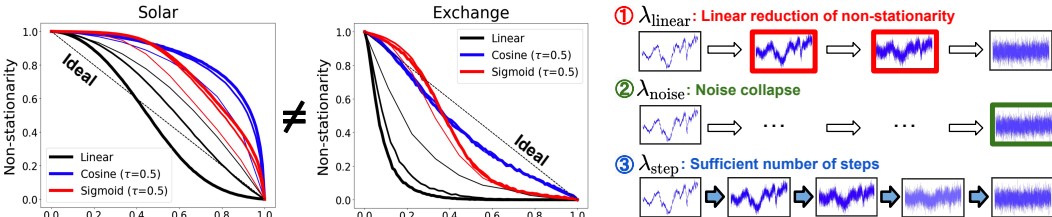

Figure 3: Non-stationarity curves of various schedules.

Figure 4: Desiderata of noise schedules.

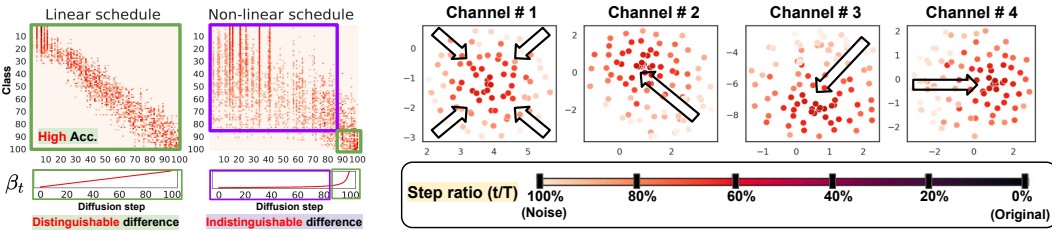

(a) Confusion matrix of proxy task.

(b) t-SNE visualizations of CNN features.

Figure 5: Proxy task classification & t-SNE visualization.

schedules yield different shapes of curves, and that the same schedule may yield different curve shapes per dataset, highlighting the importance of selecting an adaptive schedule for each dataset.

**ANT score.** To select an adaptive noise schedule for each dataset, ANT takes a dataset $\mathcal{D}$ and a schedule $\mathbf{s}$ as inputs and returns the ANT score that measures how well $\mathbf{s}$ works on $\mathcal{D}$, with a lower score indicating a better schedule. The key component of the score is the discrepancy between a linear line $l^\star$ and the non-stationarity curve $l_\mathbf{s}$, which is normalized to the range [0, 1] to account for potential differences in scales across statistics and datasets. On top of that, as shown in Figure 4, ANT considers two other desiderata of noise schedules: 1) *Noise collapse*: A noise schedule should corrupt data to random noise at the final step of the forward process to align the training and inference stages [21]. 2) *Sufficient # of steps*: A noise schedule should have a sufficiently large $T$, as increasing it generally improves sample quality [31]. To meet the desiderata, ANT opts for a schedule with relatively low non-stationarity at the final step $l_\mathbf{s}^{(T)}$ than the first step $l_\mathbf{s}^{(1)}$, and a large $T$. Then, given $\mathcal{D}$ and $\mathbf{s}$, the ANT score is defined as:

$$\text{ANT}(\mathcal{D}, \mathbf{s}) = \lambda_{\text{linear}} \cdot \lambda_{\text{noise}} \cdot \lambda_{\text{step}} \tag{5}$$

where $\lambda_{\text{linear}} = \text{d}(l^\star, \tilde{l}_\mathbf{s})$, $\lambda_{\text{noise}} = 1 + l_\mathbf{s}^{(T)}/l_\mathbf{s}^{(1)}$, and $\lambda_{\text{step}} = 1 + 1/T$ are the terms considering linear reduction of non-stationarity, noise collapse, and sufficient steps, respectively, and d is a discrepancy metric between the two lines. Among various metrics, we use the difference in the area under the curve (AUC) using the trapezoidal rule, as it empirically performs the best in our experiments. However, we note that ANT is robust across different metrics d, as shown in Table 9. The pseudocode for calculating the ANT score is described in Appendix E.

## 3.2 Diffusion Step Embedding for TS Diffusion Models

Diffusion models take the $t$-th step data ($\mathbf{x}^t$) and the step number $t$ as inputs, where $t$ is typically encoded in a DE. However, we argue that DE is *not necessary for TS diffusion models employing a linear schedule*, because information about the step is inherent in the data. To validate our claim, we conduct two experiments: 1) A proxy classification task to predict $t$ with $\mathbf{x}^t$, and 2) visualization of the embeddings of $\mathbf{x}^t$ with various $t$.

**Proxy classification.** We design a proxy classification task that identifies the step number in the forward diffusion pass given a noisy TS. For the task, we sample multiple subseries from M4 [23], each with varying steps of up to 100 steps. We build a 1D convolutional neural network (CNN) classifier with the architecture of [`Conv1D - ReLU - Flatten - Linear`], where `Conv1D` has the kernel size of 3 and 4 output channels. Figure 5a shows the results in confusion matrices, indicating that a TS corrupted using a linear schedule retains step information (high accuracy), whereas a TS corrupted using a non-linear schedule does not (low accuracy), making DE redundant for a model employing a linear schedule. We argue that this is due to the variance of noise for each step ($\beta_t$), as a non-linear schedule adds most of the noise at the end of the steps, making a TS less distinguishable for most of the steps. In contrast, a linear schedule that gradually increases $\beta_t$ makes a TS more distinguishable across steps, allowing us to eliminate the DE in the model.

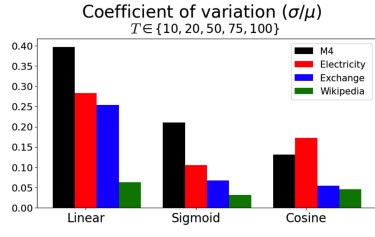
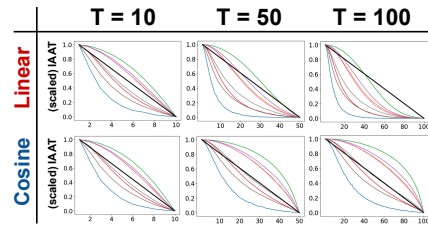

(a) Robust CRPS to $T$.  (b) Robust non-stationarity curves to $T$.

Figure 6: **Robustness of performance to** $T$. (a) shows that CRPS of non-linear schedules are robust to $T$, supported by (b) which shows the robustness of non-stationarity curves to $T$.

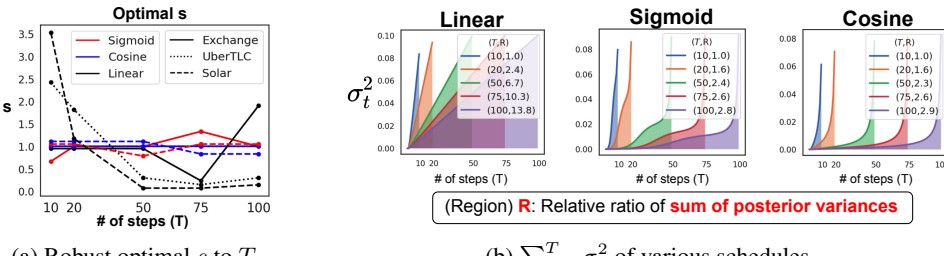

(a) Robust optimal $s$ to $T$.  (b) $\sum_{t=1}^{T} \sigma_t^2$ of various schedules.

Figure 7: **Robustness of optimal** $s$ **to** $T$. (a) shows that optimal $s$ of non-linear schedules are robust to $T$, supported by (b) which shows the robustness of sum of the posterior variances to $T$.

**t-SNE visualization.** Figure 5b depicts the t-SNE visualizations of CNN features extracted from the proxy classification model, representing a noisy TS at various steps corrupted by a linear schedule. The results show directional point movement across all output channels as the step progresses, implying that diffusion step information is inherently present in the TS. As discussed in Appendix P, t-SNE visualizations employing a non-linear schedule also show directional point movement but with a common pattern across all output channels, indicating limited step information compared to a linear schedule. Based on these observations, we remove the DE from the diffusion models when employing a linear schedule. In experiments, we observe that DE is useful for models with non-linear schedules, but not for models with linear schedules, as shown in Table 6.

### 3.3 Robustness of Non-linear Schedules

In diffusion models, the number of steps $T$ determines the efficiency of overall process and sample quality. We argue that non-linear schedules are more robust to $T$ than linear schedules in two aspects: 1) performance and 2) the optimal scale parameter $s$ controlling the self-guidance in Eq. (4).

**Robustness of performance to** $T$. Figure 6a depicts the coefficient of variation of continuous ranked probability score (CRPS) [9] of forecasting task with various $T$, illustrating that the performance is robust to $T$ when employing a non-linear schedule. This is supported by Figure 6b, which shows the non-stationarity curves of various schedules, with each color representing a different dataset. The figure indicates that the shape of the curve, which determines the ANT score, remains consistent regardless of $T$ when employing a non-linear schedule. It is important to note that the ANT score is highly correlated with the performance, as discussed in Section 4.4.

**Robustness of optimal** $s$ **to** $T$. Figure 7a shows that the optimal $s$ is sensitive to $T$ when employing linear schedules, whereas robust when using non-linear schedules. We discover that this robustness stems from the posterior variance ($\sigma_t^2$) of a schedule, which affects the self-guidance term in Eq. (4). Figure 7b illustrates the sum of the posterior variances ($\sum_{t=1}^{T} \sigma_t^2$) of various schedules, indicating that this sum is more robust to the choice of $T$ for non-linear schedules compared to linear schedules.

## 4 Experiments

**Experimental setup.** We demonstrate the effectiveness of our proposed method on three tasks: TS forecasting, refinement, and generation. For evaluation, we adopt ANT to TSDiff [15], where we use the official code to replicate the results. All experimental protocols adhere to that of TSDiff, where we utilize the CRPS [9] to assess the quality of probabilistic forecasts. We conduct experiments on eight TS datasets from different domains – Solar [17], Electricity [3], Traffic [3], Exchange [17], M4 [23], UberTLC [7], KDDCup [10], and Wikipedia [8]. We present the mean and standard deviations calculated from three independent trials.

| Method | Solar | Electricity | Traffic | Exchange | M4 | UberTLC | KDDCup | Wikipedia |
|---|---|---|---|---|---|---|---|---|
| DeepAR | $0.389_{\pm0.001}$ | $0.054_{\pm0.000}$ | $0.099_{\pm0.001}$ | $0.011_{\pm0.003}$ | $0.052_{\pm0.006}$ | $\mathbf{0.161_{\pm0.002}}$ | $0.414_{\pm0.027}$ | $0.231_{\pm0.008}$ |
| DeepState | $0.379_{\pm0.002}$ | $0.075_{\pm0.004}$ | $0.146_{\pm0.018}$ | $0.011_{\pm0.001}$ | $0.041_{\pm0.002}$ | $0.288_{\pm0.087}$ | - | $0.318_{\pm0.019}$ |
| Transformer | $0.419_{\pm0.008}$ | $0.076_{\pm0.018}$ | $\underline{0.102_{\pm0.002}}$ | $0.010_{\pm0.000}$ | $0.040_{\pm0.014}$ | $0.192_{\pm0.004}$ | $0.411_{\pm0.021}$ | $\underline{0.214_{\pm0.001}}$ |
| TFT | $0.417_{\pm0.023}$ | $0.086_{\pm0.008}$ | $0.134_{\pm0.007}$ | $\mathbf{0.007_{\pm0.000}}$ | $0.039_{\pm0.001}$ | $0.193_{\pm0.006}$ | $0.581_{\pm0.053}$ | $0.229_{\pm0.006}$ |
| CSDI | $\underline{0.352_{\pm0.005}}$ | $0.054_{\pm0.000}$ | $0.159_{\pm0.002}$ | $0.033_{\pm0.014}$ | $0.040_{\pm0.003}$ | $0.206_{\pm0.002}$ | $0.318_{\pm0.002}$ | $0.289_{\pm0.017}$ |
| TSDiff | $0.399_{\pm0.003}$ | $\underline{0.049_{\pm0.000}}$ | $0.105_{\pm0.001}$ | $0.012_{\pm0.001}$ | $\underline{0.036_{\pm0.001}}$ | $0.172_{\pm0.005}$ | $\underline{0.335_{\pm0.002}}$ | $0.221_{\pm0.001}$ |
| TSDiff+ANT | $\mathbf{0.326_{\pm0.000}}$ | $\mathbf{0.047_{\pm0.000}}$ | $\mathbf{0.101_{\pm0.000}}$ | $\underline{0.009_{\pm0.000}}$ | $\mathbf{0.026_{\pm0.000}}$ | $\underline{0.163_{\pm0.000}}$ | $\mathbf{0.325_{\pm0.000}}$ | $\mathbf{0.206_{\pm0.000}}$ |
| + Gain (%) | +22.4% | +4.1% | +3.9% | +25.1% | +27.8% | +5.2% | +3.0% | +6.8% |

(a) TS forecasting task: Standard horizon $H$.

| TSDiff | $\alpha$ | Solar | Traffic | Electricity | Exchange | M4 | UberTLC | KDDCup | Wikipedia |
|---|---|---|---|---|---|---|---|---|---|
| w/o ANT | 1/4 | - | - | $0.044_{\pm0.000}$ | $0.007_{\pm0.000}$ | $0.033_{\pm0.000}$ | $\mathbf{0.156_{\pm0.003}}$ | $0.275_{\pm0.000}$ | $0.182_{\pm0.000}$ |
| | 1/2 | $\mathbf{0.184_{\pm0.001}}$ | $0.100_{\pm0.000}$ | $0.047_{\pm0.000}$ | $0.012_{\pm0.000}$ | $0.041_{\pm0.000}$ | $\mathbf{0.157_{\pm0.002}}$ | $0.452_{\pm0.003}$ | $0.189_{\pm0.000}$ |
| | 2 | $0.359_{\pm0.001}$ | $\mathbf{0.097_{\pm0.000}}$ | $0.052_{\pm0.000}$ | $0.012_{\pm0.000}$ | $0.053_{\pm0.001}$ | $0.196_{\pm0.004}$ | $0.502_{\pm0.004}$ | $0.241_{\pm0.000}$ |
| | 4 | $0.383_{\pm0.000}$ | $0.110_{\pm0.000}$ | $0.056_{\pm0.000}$ | $\mathbf{0.015_{\pm0.000}}$ | $0.046_{\pm0.000}$ | $0.264_{\pm0.005}$ | $0.495_{\pm0.004}$ | $0.279_{\pm0.001}$ |
| w/ ANT | 1/4 | - | - | $\mathbf{0.041_{\pm0.000}}$ | $\mathbf{0.006_{\pm0.000}}$ | $\mathbf{0.025_{\pm0.000}}$ | $0.162_{\pm0.001}$ | $\mathbf{0.256_{\pm0.001}}$ | $\mathbf{0.166_{\pm0.000}}$ |
| | 1/2 | $0.218_{\pm0.000}$ | $\mathbf{0.099_{\pm0.000}}$ | $\mathbf{0.043_{\pm0.000}}$ | $\mathbf{0.009_{\pm0.000}}$ | $\mathbf{0.028_{\pm0.000}}$ | $0.162_{\pm0.001}$ | $\mathbf{0.247_{\pm0.000}}$ | $\mathbf{0.182_{\pm0.000}}$ |
| | 2 | $\mathbf{0.356_{\pm0.000}}$ | $0.102_{\pm0.000}$ | $\mathbf{0.051_{\pm0.000}}$ | $\mathbf{0.011_{\pm0.000}}$ | $\mathbf{0.040_{\pm0.000}}$ | $\mathbf{0.184_{\pm0.002}}$ | $\mathbf{0.497_{\pm0.002}}$ | $0.235_{\pm0.000}$ |
| | 4 | $\mathbf{0.376_{\pm0.000}}$ | $\mathbf{0.103_{\pm0.000}}$ | $\mathbf{0.053_{\pm0.000}}$ | $0.025_{\pm0.001}$ | $\mathbf{0.043_{\pm0.001}}$ | $\mathbf{0.259_{\pm0.003}}$ | $\mathbf{0.453_{\pm0.002}}$ | $0.275_{\pm0.000}$ |

(b) TS forecasting task: Various horizons $\alpha \cdot H$.

Table 2: Results of TS forecasting.

| TSDiff | Refinement | Solar | Electricity | Traffic | Exchange | M4 | UberTLC | KDDCup |
|---|---|---|---|---|---|---|---|---|
| w/o ANT | LMC-MS | $0.494_{\pm0.019}$ | $0.059_{\pm0.004}$ | $0.113_{\pm0.001}$ | $0.013_{\pm0.001}$ | $0.040_{\pm0.002}$ | $0.187_{\pm0.007}$ | $0.458_{\pm0.015}$ |
| | LMC-Q | $0.516_{\pm0.020}$ | $0.055_{\pm0.003}$ | $0.119_{\pm0.002}$ | $0.009_{\pm0.000}$ | $0.034_{\pm0.001}$ | $0.228_{\pm0.010}$ | $0.346_{\pm0.010}$ |
| | ML-MS | $0.503_{\pm0.016}$ | $0.063_{\pm0.005}$ | $0.117_{\pm0.002}$ | $0.015_{\pm0.002}$ | $0.045_{\pm0.003}$ | $0.203_{\pm0.007}$ | $0.472_{\pm0.015}$ |
| | ML-Q | $0.523_{\pm0.021}$ | $0.056_{\pm0.003}$ | $0.121_{\pm0.003}$ | $0.010_{\pm0.001}$ | $0.032_{\pm0.001}$ | $0.240_{\pm0.010}$ | $0.350_{\pm0.011}$ |
| | Avg. | 0.509 | 0.058 | 0.118 | 0.012 | 0.038 | 0.215 | 0.407 |
| w/ ANT | LMC-MS | $\mathbf{0.447_{\pm0.012}}$ | $\mathbf{0.055_{\pm0.000}}$ | $\mathbf{0.113_{\pm0.000}}$ | $\mathbf{0.009_{\pm0.000}}$ | $\mathbf{0.035_{\pm0.000}}$ | $\mathbf{0.180_{\pm0.001}}$ | $\mathbf{0.410_{\pm0.016}}$ |
| | LMC-Q | $\mathbf{0.495_{\pm0.005}}$ | $\mathbf{0.054_{\pm0.000}}$ | $\mathbf{0.115_{\pm0.000}}$ | $\mathbf{0.009_{\pm0.000}}$ | $\mathbf{0.033_{\pm0.000}}$ | $\mathbf{0.186_{\pm0.001}}$ | $\mathbf{0.364_{\pm0.021}}$ |
| | ML-MS | $\mathbf{0.449_{\pm0.005}}$ | $\mathbf{0.058_{\pm0.000}}$ | $\mathbf{0.115_{\pm0.000}}$ | $\mathbf{0.010_{\pm0.000}}$ | $\mathbf{0.037_{\pm0.000}}$ | $\mathbf{0.182_{\pm0.001}}$ | $\mathbf{0.419_{\pm0.015}}$ |
| | ML-Q | $\mathbf{0.499_{\pm0.006}}$ | $\mathbf{0.056_{\pm0.000}}$ | $\mathbf{0.115_{\pm0.001}}$ | $\mathbf{0.010_{\pm0.000}}$ | $\mathbf{0.034_{\pm0.000}}$ | $\mathbf{0.187_{\pm0.001}}$ | $\mathbf{0.367_{\pm0.021}}$ |
| | Avg. | **0.472** | **0.056** | **0.115** | **0.009** | **0.035** | **0.184** | **0.389** |
| + Gain (%) | | +7.3% | +3.5% | +2.6% | +25.0% | +7.9% | +14.4% | +4.4% |

Table 3: Results of TS refinement.

**Noise schedules.** ANT finds the best schedule for a given dataset from candidate schedules based on the ANT score. Table 1 shows candidate schedules that we use for our experiments based on the three components: noise function ($f$), temperature ($\tau$), and the number of diffusion steps ($T$). We denote $f(T)$ for a linear schedule and $f(T, \tau)$ for a non-linear schedule. Note that TSDiff [15] employs Lin(100) as a common schedule for all datasets, following TimeGrad [26].

| | 1) $f$ | 2) $\tau$ | 3) $T$ |
|---|---|---|---|
| Linear (Lin) | | - | [10,20,50,75,100] |
| Cosine (Cos) | | [0.5,1.0,2.0] | [10,20,50,75,100] |
| Sigmoid (Sig) | | [0.3,0.5,1.0] | [10,20,50,75,100] |

Table 1: Candidate schedules.

## 4.1 Time Series Forecasting

For baseline methods in forecasting tasks, we compare our method with various approaches, including DeepAR [28], MQ-CNN [36], DeepState [25], Transformer [35], TFT [20], CSDI [34], and TSDiff [15]. We note that we do not aim to outperform all SOTA methods, but rather to demonstrate the efficacy of our proposed adaptive noise schedules when applied within the framework of TS diffusion models. Nonetheless, Table 2a indicates that TSDiff employing the noise schedule proposed by our method achieves the best performance on all datasets.

To show the effectiveness of our method across various prediction horizons, we conduct an experiment with different horizons using the same input window. Specifically, we multiply $\alpha$ by the standard horizon $H$ used in Table 2a, where the range of $\alpha$ depends on the sequence length of the dataset. Table 2b shows that ANT enhances TSDiff with variable prediction lengths.

## 4.2 Time Series Refinement

We conduct an experiment refining the forecasting results of other base forecasters to evaluate the quality of the implicit probability density of TSDiff applied with ANT. For the experiment, we employ Gaussian (MS) and asymmetric Laplace negative log-likelihoods (Q) as regularizers for both energy (LMC) and maximum likelihood (ML)-based refinement, following TSDiff. Table 3 displays the results using a linear model as the base forecaster, showing that TSDiff, when applied with our method, yields performance gains across all datasets and refinement methods.

| | Generator | Solar | Traffic | Exchange | M4 | UberTLC | KDDCup | Wikipedia |
|---|---|---|---|---|---|---|---|---|
| **Linear** | TimeVAE | $0.933_{\pm0.147}$ | $0.236_{\pm0.010}$ | $0.024_{\pm0.004}$ | $0.074_{\pm0.003}$ | $0.354_{\pm0.020}$ | $1.020_{\pm0.179}$ | $0.643_{\pm0.068}$ |
| | TimeGAN | $1.140_{\pm0.583}$ | $0.398_{\pm0.092}$ | $0.011_{\pm0.000}$ | $0.140_{\pm0.053}$ | $0.665_{\pm0.104}$ | $0.713_{\pm0.009}$ | $0.421_{\pm0.023}$ |
| | TSDiff | $0.597_{\pm0.005}$ | $*0.181_{\pm0.000}$ | $0.012_{\pm0.001}$ | $0.045_{\pm0.007}$ | $0.291_{\pm0.084}$ | $0.504_{\pm0.009}$ | $0.392_{\pm0.013}$ |
| | TSDiff+ANT | $0.533_{\pm0.000}$ | $0.177_{\pm0.001}$ | $0.012_{\pm0.000}$ | $0.040_{\pm0.001}$ | $0.232_{\pm0.005}$ | $0.426_{\pm0.006}$ | $0.350_{\pm0.005}$ |
| | + Gain (%) | +10.7% | +2.2% | +0.0% | +11.1% | +20.3% | +15.5% | +10.7% |
| **DeepAR** | TimeVAE | $0.493_{\pm0.012}$ | $0.155_{\pm0.006}$ | $0.009_{\pm0.000}$ | $0.039_{\pm0.010}$ | $0.278_{\pm0.009}$ | $0.621_{\pm0.003}$ | $0.440_{\pm0.012}$ |
| | TimeGAN | $0.976_{\pm0.739}$ | $0.419_{\pm0.122}$ | $0.008_{\pm0.001}$ | $0.121_{\pm0.035}$ | $0.594_{\pm0.125}$ | $0.690_{\pm0.091}$ | $0.322_{\pm0.048}$ |
| | TSDiff | $0.478_{\pm0.007}$ | $*0.140_{\pm0.003}$ | $0.029_{\pm0.017}$ | $0.042_{\pm0.024}$ | $0.268_{\pm0.007}$ | $*0.416_{\pm0.028}$ | $0.225_{\pm0.002}$ |
| | TSDiff+ANT | $0.424_{\pm0.005}$ | $0.139_{\pm0.001}$ | $0.013_{\pm0.000}$ | $0.033_{\pm0.000}$ | $0.185_{\pm0.001}$ | $0.343_{\pm0.012}$ | $0.240_{\pm0.001}$ |
| | + Gain (%) | +11.3% | +0.7% | +55.2% | +21.4% | +30.1% | +17.5% | −6.7% |
| **Transformer** | TimeVAE | $0.520_{\pm0.030}$ | $0.163_{\pm0.018}$ | $0.011_{\pm0.001}$ | $0.035_{\pm0.011}$ | $0.291_{\pm0.008}$ | $0.717_{\pm0.181}$ | $0.451_{\pm0.017}$ |
| | TimeGAN | $0.972_{\pm0.687}$ | $0.413_{\pm0.2044}$ | $0.009_{\pm0.001}$ | $0.114_{\pm0.052}$ | $0.685_{\pm0.448}$ | $0.632_{\pm0.016}$ | $0.314_{\pm0.045}$ |
| | TSDiff | $0.457_{\pm0.008}$ | $*0.153_{\pm0.003}$ | $0.031_{\pm0.009}$ | $0.069_{\pm0.003}$ | $0.272_{\pm0.016}$ | $0.343_{\pm0.014}$ | $0.239_{\pm0.010}$ |
| | TSDiff+ANT | $0.444_{\pm0.002}$ | $0.148_{\pm0.004}$ | $0.014_{\pm0.000}$ | $0.039_{\pm0.001}$ | $0.225_{\pm0.003}$ | $0.366_{\pm0.008}$ | $0.234_{\pm0.001}$ |
| | + Gain (%) | +2.8% | +3.3% | +54.8% | +43.5% | +17.3% | −6.6% | +2.1% |

Table 4: Results of TS generation.

| | | | Solar | Electricity | Traffic | Exchange | M4 | UberTLC | KDDCup | Wikipedia | Avg. |
|---|---|---|---|---|---|---|---|---|---|---|---|
| **Schedule** | TSDiff | w/o ANT | | | | Lin(100) | | | | | - |
| | | w/ ANT | Lin(100) | Cos(75,2.0) | Lin(50) | Cos(50,0.5) | Cos(100,1.0) | Lin(20) | Lin(50) | Cos(75,2.0) | |
| | Oracle | | | | Cos(75,2.0) | | | | | | |
| **CRPS** | TSDiff | w/o ANT | 0.399 | 0.049 | 0.105 | 0.012 | 0.036 | 0.172 | 0.335 | 0.221 | 0.166 |
| | | w/ ANT | 0.326 | 0.047 | 0.101 | 0.009 | 0.026 | 0.163 | 0.325 | 0.206 | 0.150 |
| | Oracle | | 0.326 | 0.047 | 0.099 | 0.009 | 0.026 | 0.163 | 0.325 | 0.206 | 0.150 |

Table 5: Schedules proposed by ANT and forecasting results with them.

## 4.3 Time Series Generation

To assess the generation quality of TS diffusion models with ANT, we evaluate the forecasting performance of downstream forecasters trained on the synthetic samples generated by various models. For the baseline methods, we employ TimeGAN [39], TimeVAE [5], and TSDiff [15]. For the downstream models, we employ a simple linear (ridge) regression model, DeepAR [28], and Transformer [35]. The results are shown in Table 4, indicating that ANT improves the performance of TSDiff, leading to SOTA performance in most cases. Further details regarding the generation task with Electricity are discussed in Section O.

## 4.4 Analysis

**Proposed noise schedule.** Table 5 shows the schedules proposed by ANT and the schedules that yield the best performance (oracle) among all candidate schedules, along with the forecasting performance obtained when using these schedules. The result indicates that the schedules selected by ANT are mostly consistent with the oracle, and even for an exceptional case that ANT fails to select the best schedule, e.g., on Traffic, the performance gap from the oracle is not significant.

**Necessity of DE.** Table 6 displays the forecasting results of TSDiff applied with ANT, both with and without the use of DE across eight datasets. The result indicates that for all datasets except for Traffic, models trained with a linear schedule perform better when trained without DE, while models trained with a non-linear schedule perform better when trained with DE. Based on these findings, we do not use DE when employing a linear schedule.

| Dataset | | | | TSDiff+ANT | | TSDiff |
|---|---|---|---|---|---|---|
| | $f$ | $\tau$ | $T$ | w/o DE | w/ DE | |
| UberTLC | Linear | - | 20 | **0.163** | 0.169 | 0.172 |
| KDDCup | | - | 50 | **0.325** | 0.340 | 0.335 |
| Traffic | | - | 50 | 0.101 | **0.098** | 0.105 |
| Solar | | - | 100 | **0.326** | 0.399 | 0.399 |
| Exchange | Cosine | 0.5 | 50 | 0.010 | **0.009** | 0.012 |
| Electricity | | 2.0 | 75 | 0.050 | **0.047** | 0.049 |
| Wikipedia | | 2.0 | 75 | **0.206** | 0.206 | 0.221 |
| M4 | | 1.0 | 100 | 0.052 | **0.026** | 0.036 |

Table 6: TS forecasting w/ & w/o DE.

**Ablation study.** To show the effectiveness of each component in ANT, we conduct an ablation study in Table 7. The result demonstrates that enforcing the schedule to gradually drop the non-stationarity is the most important component ($\lambda_{\text{linear}}$), and ensuring noise collapse ($\lambda_{\text{noise}}$) and sufficient steps ($\lambda_{\text{step}}$) also help ANT achieve oracle for three datasets [23, 17, 3] when combined together. Figure 8 shows the correlation between the ANT score and CRPS on forcasting tasks via a scatter plot of candidate schedules and a bar plot of schedules selected in Table 7, implying that there is a strong correlation between the ANT score and CRPS, and the components in ANT strengthen the correlation further.

| ANT components | | | Solar | Electricity | Traffic | Exchange | M4 | UberTLC | KDDCup | Wikipedia |
|---|---|---|---|---|---|---|---|---|---|---|
| $\lambda_{\text{linear}}$ | $\lambda_{\text{noise}}$ | $\lambda_{\text{step}}$ | | | | | | | | |
| ✓ | | | Lin(75) | Cos(10,1.0) | Lin(20) | **Cos(50,0.5)** | Cos(10,0.5) | **Lin(20)** | **Lin(50)** | **Cos(75,2.0)** |
| ✓ | ✓ | | Lin(75) | Cos(10,1.0) | Lin(20) | **Cos(50,0.5)** | Cos(10,0.5) | **Lin(20)** | **Lin(50)** | **Cos(75,2.0)** |
| ✓ | | ✓ | Lin(75) | Cos(10,1.0) | Lin(20) | **Cos(50,0.5)** | Cos(10,0.5) | **Lin(20)** | **Lin(50)** | **Cos(75,2.0)** |
| ✓ | ✓ | ✓ | **Lin(100)** | **Cos(75,2.0)** | Lin(50) | **Cos(50,0.5)** | **Cos(100,1.0)** | **Lin(20)** | **Lin(50)** | **Cos(75,2.0)** |
| Oracle | | | **Lin(100)** | **Cos(75,2.0)** | **Cos(75,2.0)** | **Cos(50,0.5)** | **Cos(100,1.0)** | **Lin(20)** | **Lin(50)** | **Cos(75,2.0)** |

Table 7: **Ablation study.** The table shows the schedules proposed by ANT when using some or all components of the ANT score.

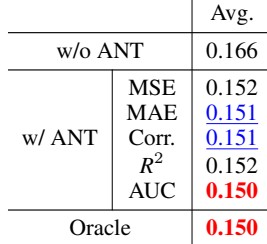
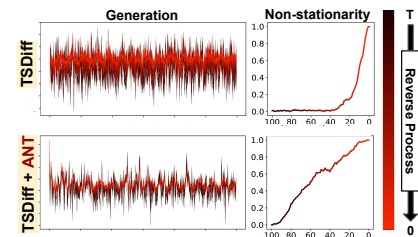

(a) ANT score vs. CRPS.   (b) Correlation between ANT score and CRPS.

Figure 8: **Ablation study.** (a) depicts the scatter plot of ANT score and forecasting result of various schedules, where a schedule with a lower score yields better performance in general. (b) shows that using all three components of ANT score results in the highest correlation.

| | | Avg. |
|---|---|---|
| w/o ANT | | 0.166 |
| | VarAC | 0.158 |
| w/ ANT | Lag1AC | 0.151 |
| | IAAT | **0.150** |
| Oracle | | **0.150** |

Table 8: Robustness to statistics.

| | | Avg. |
|---|---|---|
| w/o ANT | | 0.166 |
| | MSE | 0.152 |
| | MAE | 0.151 |
| w/ ANT | Corr. | 0.151 |
| | $R^2$ | 0.152 |
| | AUC | **0.150** |
| Oracle | | **0.150** |

Table 9: Robustness to d.

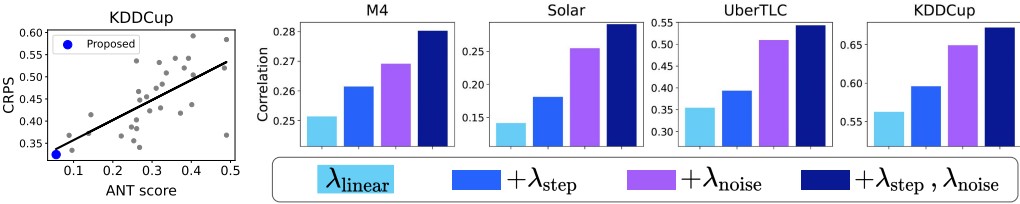

Figure 9: Generation process.

**Robustness to statistics.** To see if ANT is sensitive to the choice of statistics of non-stationarity, we compare IAAT with other statistics including Lag1AC and VarAC. Table 8 shows the average CRPS across eight datasets in forecasting tasks, demonstrating that ANT with different statistics still outperforms the model without ANT, while IAAT performs best. We conjecture that this is because IAAT considers AC at all time lags, whereas Lag1AC only considers a single lag and VarAC focuses on the variance of AC rather than their exact values.

**Robustness to d.** Among the various metrics that can be employed for d to measure the discrepancy between the two lines, we compare AUC with the mean-squared error (MSE), mean-absolute error (MAE), Pearson correlation (Corr.), and R-squared ($R^2$). Table 9 presents the average CRPS of eight datasets on the forecasting task, demonstrating the robustness of our method to the choice of metric.

**Generation process.** Figure 9 illustrates the generation process of a TSDiff model trained on M4 with and without ANT, where the color changes from black to red as the step progresses. The figure on the left shows the generated TS for each step, while the one on the right shows the non-stationarity of each step in terms of IAAT. As shown in Figure 9, ANT indeed makes the generation process linearly increase the non-stationarity.

**Effect of $T$ on performance.** While it was believed that the larger the $T$, the better the performance in diffusion models [31], we argue that this is not always true in the TS domain. Table 10 shows the oracle performance for each $T$ among candidate schedules on forecasting tasks, indicating that the best performance is not always achieved with the largest $T$, e.g., the best performance is achieved with $T = 20$ on UberTLC [7]. Furthermore, as shown in Figure 10, ANT often outperforms the baseline method TSDiff with smaller $T$, achieving the oracle performance in most cases.

**Efficiency of ANT.** ANT offers efficiency in two aspects: 1) Statistics of non-stationarity can be precomputed offline with a small additional cost, and 2) the proposed schedule with $T$ smaller than that of the base schedule results in faster inference. First, Table 11a displays the time spent calculating IAAT using the entire train dataset of Traffic [3] with schedules of five different values for $T$. The results indicate that the computational cost is negligible compared to the training time shown in Table 11b, making ANT practical for use. Second, Table 11b shows the training and inference time on Traffic, where we report the training time with 1000 epochs and the inference time for a TS data averaged on the test dataset. The results demonstrate that we can save up to 50% of the inference time and reduce the training time by eliminating the DE.

| | $T$ | Solar | Electricity | Traffic | Exchange | M4 | UberTLC | KDDCup | Wikipedia |
|---|---|---|---|---|---|---|---|---|---|
| TSDiff | | 0.399 | 0.049 | 0.105 | 0.012 | 0.036 | 0.172 | 0.335 | 0.221 |
| TSDiff+ANT | | **0.326** | **0.047** | 0.101 | **0.009** | **0.026** | **0.163** | **0.325** | **0.206** |
| Oracle | 10 | 0.371 | 0.062 | 0.127 | 0.011 | 0.039 | 0.190 | 0.387 | 0.225 |
| | 20 | 0.342 | 0.050 | 0.103 | 0.010 | 0.031 | **0.163** | 0.366 | 0.210 |
| | 50 | 0.330 | 0.049 | 0.100 | 0.009 | 0.027 | 0.165 | **0.325** | 0.207 |
| | 75 | 0.351 | **0.047** | **0.099** | 0.010 | 0.028 | 0.169 | 0.334 | **0.206** |
| | 100 | **0.326** | 0.048 | 0.101 | 0.010 | **0.026** | 0.169 | 0.368 | 0.208 |

Table 10: Oracle (lowest CRPS) under fixed $T$.

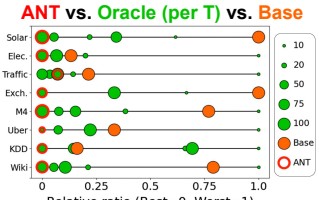

Figure 10: ANT vs. Oracle.

| Dataset: Traffic | | | | | |
|---|---|---|---|---|---|
| $T$ | 10 | 20 | 50 | 75 | 100 |
| Time | 11.1 | 22.6 | 60.2 | 93.2 | 120.0 |

(a) IAAT calculation time (sec).

| Dataset: Traffic | | | |
|---|---|---|---|
| Train (min) | | Inference (sec) | |
| w/o ANT | w/ ANT | w/o ANT | w/ ANT |
| 100.5 | **92.0** | 1.28 | **0.64** |

(b) Train & inference time.

| | | $T$ | Avg. |
|---|---|---|---|
| w/o ANT | | 100 | 0.166 |
| w/ ANT | w/ con. | $\leq 50$ | 0.157 |
| | w/o con. | $\leq 100$ | **0.150** |

(c) With constraint on $T$.

Table 11: Efficiency of ANT.

| | UberTLC | | Traffic | |
|---|---|---|---|---|
| | w/o Cos* | w/ Cos* | w/o Cos* | w/ Cos* |
| Schedule | Lin(20) | Cos*(20,0.2) | Lin(50) | Cos*(20,2.0) |
| ANT score | 0.054 | **0.053** | 0.088 | **0.050** |
| CRPS | 0.163 | **0.161** | 0.101 | **0.099** |

Table 12: Flexibility of ANT.

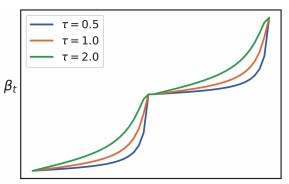

Figure 11: $\beta_t$ of Cos*.

Furthermore, ANT offers a performance-computational cost trade-off in that it can be optimized with a constraint on $T$ by selecting a schedule among candidates with $T$ no larger than $T^*$ ($T \leq T^*$). Table 11c shows the forecasting results with a constraint of $T^* = 50$, achieving better performance with no more than half of the steps of the base schedule. Further analyses regarding computational time and forecasting results under constraints on $T$ are discussed in Appendix N and J, respectively.

**Flexibility of ANT.** As ANT does not require direct access to the noise schedule but the TS data corrupted by the schedule, any noise schedule can be a candidate. To confirm if ANT can judge the usefulness of non-trivial noise schedules other than the functions described in Table 1, we experiment an ensemble of cosine functions (Cos*), illustrated in Figure 11. Table 12 shows the results on two datasets [7, 3] that achieve lower ANT scores with Cos*, indicating the potential for a better schedule to be selected by relaxing structural constraints, or even without any functional form $f$.

**Comparison with other schedules.** To demonstrate the effectiveness of ANT compared to noise schedules from other domains, we compare ANT with a cosine schedule (Cos) [24] and a schedule that enforces zero terminal SNR (Zero) [21], applied to TSDiff. Table 13 shows the average CRPS across eight datasets on forecasting tasks, where ANT outperforms the others. The results highlight the importance of a schedule being adaptive to the given TS dataset and considering the corruption speed.

| Schedule | Avg. |
|---|---|
| Linear | 0.166 |
| Cos [24] | 0.160 |
| Zero [21] | 0.185 |
| ANT | **0.150** |

Table 13: Comparison with other schedules.

## 5 Conclusion

In this work, we introduce ANT, a method designed to select an adaptive noise schedule for TS diffusion models using the statistics of non-stationarity. ANT is practical in that it predetermines a noise schedule before training, as the statistics can be precomputed offline with minimal additional cost. The proposed ANT is a simple yet effective method for TS diffusion models across various tasks, e.g., ANT improves the SOTA method TSDiff by 9.5% on average across eight forecasting tasks, highlighting the importance of adaptive schedules tailored to the characteristics of TS datasets.

**Limitations and future work.** We note that our contribution is not in finding the optimal schedule but in proposing a criterion for efficiently selecting a better noise schedule from a set of candidates based on the dataset characteristics. While our experiments search for an appropriate schedule from 35 candidates derived from linear, cosine, and sigmoid functions, any schedule can be considered a candidate for ANT, and incorporating additional schedules could lead to further performance gains at the expense of longer search times. We leave developing algorithms to determine the optimal schedule parameters based on our proposed criterion for future work. We hope that our research motivates further research across various domains to incorporate domain knowledge in the design of machine learning models, extending beyond noise schedules for diffusion models.

## Acknowledgements

This work was supported by the National Research Foundation of Korea (NRF) grant funded by the Korea government (MSIT) (2020R1A2C1A01005949, 2022R1A4A1033384, RS-2023-00217705, RS-2024-00341749), the MSIT(Ministry of Science and ICT), Korea, under the ICAN(ICT Challenge and Advanced Network of HRD) support program (RS-2023-00259934) supervised by the IITP(Institute for Information & Communications Technology Planning & Evaluation), the Yonsei University Research Fund (2024-22-0148), and the Son Jiho Research Grant of Yonsei University (2023-22-0006).

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

## A  Dataset Description

In our experiments, we employ eight widely-used univariate datasets from various fields which can be found in GluonTS [2] in their preprocessed form, with training and test splits provided. Following TSDiff [15], the validation set is created by splitting a portion of the training dataset, with the split ratio determined by the sizes of the training and test datasets. Table A.1 shows the statistics of the dataset, annotated with their corresponding frequencies (daily or hourly) and lengths of predictions. For the daily datasets, the sequence lengths are set at 390, roughly equivalent to 13 months. For the hourly datasets, we utilize sequence lengths of 360, corresponding to 15 days. These sequences are created by randomly slicing the original time series at various time steps. Below, we provide a brief description of each dataset.

- **Solar** [17]: This dataset contains data on solar power production in 2006 from 137 solar power plants located in Alabama. [2]
- **Electricity** [17, 3]: This dataset includes electricity consumption data from 370 customers. [3]
- **Traffic** [17, 3]: This dataset features hourly occupancy rates on the freeways in the San Francisco Bay area from 2015 to 2016. [4]
- **Exchange** [17]: This data is comprised of daily exchange rates from eight countries—Australia, the UK, Canada, Switzerland, China, Japan, New Zealand, and Singapore—from 1990 to 2016. [5]
- **M4** [23]: This is an hourly data subset derived from the M4 forecasting competition. [6]
- **UberTLC** [7]: This dataset includes data on Uber pickups in NYC from January to June 2015, collected by the NYC Taxi & Limousine Commission. [7]
- **KDDCup** [10]: This dataset consists of air quality indices (AQIs) for Beijing and London, used in the KDD Cup 2018. [8]
- **Wikipedia** [8]: This dataset contains daily number of hits on 2000 Wikipedia pages. [9]

| Dataset | $N_{\text{train}}$ | $N_{\text{test}}$ | Domain | Freq. | Median Length | $L$ | $H$ |
|---|---|---|---|---|---|---|---|
| Solar | 137 | 959 | $\mathbb{R}^+$ | H | 7009 | 336 | 24 |
| Electricity | 370 | 2590 | $\mathbb{R}^+$ | H | 5833 | 336 | 24 |
| Traffic | 963 | 6741 | $(0, 1)$ | H | 4001 | 336 | 24 |
| Exchange | 8 | 40 | $\mathbb{R}^+$ | D | 6071 | 360 | 30 |
| M4 | 414 | 414 | $\mathbb{N}$ | H | 960 | 312 | 48 |
| UberTLC | 262 | 262 | $\mathbb{N}$ | H | 4320 | 336 | 24 |
| KDDCup | 270 | 270 | $\mathbb{N}$ | H | 10850 | 312 | 48 |
| Wikipedia | 2000 | 10000 | $\mathbb{N}$ | D | 792 | 360 | 30 |

Table A.1: Statistics of datasets.

---

[2]Solar: https://www.nrel.gov/grid/solar-power-data.html

[3]Electricity: https://archive.ics.uci.edu/ml/datasets/ElectricityLoadDiagrams20112014

[4]Traffic: https://zenodo.org/record/4656132

[5]Exchange: https://github.com/laiguokun/multivariate-time-series-data

[6]M4: https://github.com/Mcompetitions/M4-methods/tree/master/Dataset

[7]UberTLC: https://github.com/fivethirtyeight/uber-tlc-foil-response

[8]KDDCup: https://zenodo.org/record/4656756

[9]Wikipedia: https://github.com/mbohlkeschneider/gluon-ts/tree/mv_release/datasets

## B   Experimental Settings

**Metrics.** Following TSDiff, we employ the Continuous Ranked Probability Score (CRPS) as our evaluation metric. It is defined by the integral of the pinball loss over the interval from 0 to 1:

$$\text{CRPS}(F^{-1}, y) = \int_0^1 2\Lambda_\kappa(F^{-1}(\kappa), y)d\kappa, \tag{B.1}$$

where $\Lambda_\kappa(q, y) = (\kappa - \mathbb{1}_{y<q})(y - q)$ denotes the pinball loss for a given quantile level $\kappa$, $F^{-1}$ denotes the predicted inverse cumulative distribution function, and $y$ denotes the observed value. Since the quantile function is not analytically accessible in practice, it is approximated using discrete quantile levels derived from sample forecasts. For the experiment, we utilize the CRPS implementation in GluonTS [2], which uses 100 sample forecasts to calculate nine predefined quantile levels $\{0.1, 0.2, 0.3, 0.4, 0.5, 0.6, 0.7, 0.8, 0.9\}$.

**Hyperparameters.** Table B.1 presents the hyperparameters of the backbone model utilized in our experiment, which are aligned with those used in TSDiff. Note that the diffusion step embeddings [35] are only applied to the model employing a non-linear schedule. Moreover, we employ skip connections for certain datasets, following the TSDiff approach, which results in improvements in validation performance. For the normalization method, we employ the mean scaler from GluonTS [2], which normalizes each TS by dividing it by the mean of the absolute values of its observations. The scale parameter $s$ which controls the self-guidance term is searched over the form $2^\alpha$ where $\alpha$ ranges through the set $[-1, 0, 1, 2, 3, 4, 5, 6, 7]$.

| Hyperparameter | | Value |
|---|---|---|
| Learning rate | | 1e-3 |
| Optimizer | | Adam |
| Batch size | | 64 |
| Epochs | | 1000 |
| Grad. clipping threshold | | 0.5 |
| Residual layers/channels | | 3/64 |
| Dim of DE | | 128 |
| Normalization | | Mean-scaling |
| Self-guidance | | Quantile |
| scale $s$ | Solar | 4 |
| | Electricity | 8 |
| | Traffic | 4 |
| | Exchange | 32 |
| | M4 | 16 |
| | UberTLC | 16 |
| | KDDCup | 4 |
| | Wikipedia | 8 |

Table B.1: Hyperparameters.

## C  Baseline Methods

For the baseline methods in TS forecasting and generation tasks, we adhere to the methods outlined in TSDiff [15].

### C.1  Methods for TS Forecasting

- **DeepAR** [28] is a probabilistic forecasting method that outputs the parameters for distributions, optimized with the (conditional) log-likelihood. The model employs an RNN-based autoregressive framework that leverages past lags and external features to predict future values.

- **DeepState** [25] integrates RNNs with linear dynamical systems (LDS), where the LDS parameters, such as transition and emission matrices, are predefined to reflect trend and seasonality of TS. The parameters of LDS and RNN are simultaneously optimized using maximum likelihood estimation.

- **Transformer** utilizes a self-attention mechanism [35] for sequence-to-sequence forecasting, employing TS lags and additional covariates to predict future distribution parameters.

- **TFT** [20] is a Transformer-based forecasting method that incorporates that employs a variable selection mechanism to filter out irrelevant features while training on quantile loss.

- **CSDI** [34] is a conditional diffusion model that adapts the DiffWave framework [16] to multivariate TS forecasting and imputation, incorporating transformer layers into its architecture.

- **TSDiff** [15] is unconditional diffusion model that utilizes a self-guidance mechanism, that leverages conditioning information during the inference stage with an unconditional network, without requiring additional architectural changes.

### C.2  Methods for TS Generation

- **TimeVAE** utilizes variational autoencoders [14], with its architecture tailored for TS analysis to handle trends and seasonality in TS.

- **TimeGAN** utilizes an autoencoder framework to train a generative adversarial network (GAN) [11] in the latent space, which is optimized with a composite loss function that integrates supervised, unsupervised, and discriminative losses.

## D  TS Diffusion Models

The goal of TS prediction tasks is to predict $\mathbf{x}_{\text{miss}}^0 \in \mathbb{R}^{d \times H}$ given $\mathbf{x}_{\text{obs}}^0 \in \mathbb{R}^{d \times L}$, where $d$, $H$, $L$ are the number of variables, the length of the target window, and the length of the input window, respectively. Most of TS diffusion models for prediction tasks are conditional models [26, 38, 18, 30, 1, 6, 29, 40, 19], employing a conditioning network $\mathcal{F}$ to construct a condition $\mathbf{c}$ using $\mathbf{x}_{\text{obs}}^0$, which contains the information of the observed values used to predict the unobserved values. The distribution for conditional TS diffusion models is formulated as

$$p_\theta\left(\mathbf{x}_{\text{miss}}^{0:T} \mid \mathbf{c}\right) = p_\theta\left(\mathbf{x}_{\text{miss}}^T\right) \prod_{t=1}^{T} p_\theta\left(\mathbf{x}_{\text{miss}}^{t-1} \mid \mathbf{x}_{\text{miss}}^t, \mathbf{c}\right), \tag{D.1}$$

where $\mathbf{c} = \mathcal{F}\left(\mathbf{x}_{\text{obs}}^0\right)$ and $\mathbf{x}_{\text{miss}}^T \sim \mathcal{N}(\mathbf{0}, \mathbf{I})$. The corresponding backward process can be expressed as

$$p_\theta\left(\mathbf{x}_{\text{miss}}^{t-1} \mid \mathbf{x}_{\text{miss}}^t, \mathbf{c}\right) = \mathcal{N}\left(\mathbf{x}_{\text{miss}}^{t-1}; \mu_\theta\left(\mathbf{x}_{\text{miss}}^t, t \mid \mathbf{c}\right), \sigma_t^2 \mathbf{I}\right), \quad t = T, \ldots, 1. \tag{D.2}$$

# E Pseudocode for ANT Score

Algorithm 1 shows the pseudocode for calculating the ANT score, which evaluates the suitability of schedules for a given dataset.

---
**Algorithm 1** Calculation of ANT score

---
**Input:** Dataset $\mathcal{D} = \{\mathbf{x}_i\}_{i=1}^N$, Schedule $\mathbf{s}$ = (schedule function: $f$, temperature: $\tau$, number of steps: $T$)

1: **for** $i = 1$ to $N$ **in parallel do**
2: $\quad$ $\mathbf{x}_i^0 := \mathbf{x}_i$
3: $\quad$ **for** $t = 1$ to $T$ **do**
4: $\quad\quad$ Add noise: $\mathbf{x}_i^t \sim q\left(\mathbf{x}_i^t \mid \mathbf{x}_i^{t-1}\right)$ based on $\mathbf{s}$
5: $\quad\quad$ Calculate non-stationarity: $\boldsymbol{g}_i^t = \mathbf{G}(\mathbf{x}_i^t)$, where $\mathbf{G}$ is a non-stationarity metric
6: $\quad$ **end for**
7: **end for**
8: Define non-stationarity curve: $l_\mathbf{s} = [l_\mathbf{s}^{(1)}, \cdots, l_\mathbf{s}^{(T)}]$, where $l_\mathbf{s}^{(t)} = \sum_{i=1}^N \boldsymbol{g}_i^t$
9: Normalize non-stationarity curve: $\tilde{l}_\mathbf{s} = \text{MinMaxScale}(l_\mathbf{s})$
10: Calculate ANT score: $\text{ANT}(\mathcal{D}, \mathbf{s}) = \lambda_\text{linear} \cdot \lambda_\text{noise} \cdot \lambda_\text{step}$,
$\quad\quad$ where $\lambda_\text{linear} = \text{d}(l^\star, \tilde{l}_\mathbf{s})$, $\lambda_\text{noise} = 1 + l_\mathbf{s}^{(T)}/l_\mathbf{s}^{(1)}$, $\lambda_\text{step} = 1 + 1/T$, $l^\star = \text{Linspace}(1, 0, T)$

**Output:** ANT score: $\text{ANT}(\mathcal{D}, \mathbf{s})$

---

# F TS Refinement Methods

TSDiff offers a strategy to improve predictions from base forecasters, adaptable to any type of base forecaster as it only requires their initial outputs. These initial forecasts are iteratively enhanced through an implicit density learned by the diffusion model, which acts as a prior. The refinement process can be conducted using one of two methods: (a) sampling from an energy function (LMC), or (b) maximizing the likelihood to determine the most probable sequence (ML). Furthermore, the approach includes an option between two types of regularizers: (a) Gaussian (MS) and (b) Laplace negative log-likelihoods (Q).

**Energy based sampling (LMC).** LMC improves the initial forecast of a given base forecaster $g$, by framing the refinement as a task of sampling from a regularized energy-based model (EBM). This can be expressed as

$$E_\theta(\mathbf{y}; \tilde{\mathbf{y}}) = -\log p_\theta(\mathbf{y}) + \lambda \mathcal{R}(\mathbf{y}, \tilde{\mathbf{y}}), \tag{F.1}$$

where $\tilde{\mathbf{y}}$ represents the time series formed by combining $\mathbf{y}_\text{obs}$ with $g(\mathbf{y}_\text{obs})$, $\mathcal{R}$ denotes a regularizer, and $\lambda$ is a Lagrange multiplier controlling the strength of regularization.

To sample from the specified EBM, overdamped Langevin Monte Carlo (LMC) [33] is used, with $\mathbf{y}_{(0)}$ initialized to $\tilde{\mathbf{y}}$. The iterative refinement process is outlined as follows:

$$\mathbf{y}_{(i+1)} = \mathbf{y}_{(i)} - \eta \nabla_{\mathbf{y}_{(i)}} E_\theta\left(\mathbf{y}_{(i)}; \tilde{\mathbf{y}}\right) + \sqrt{2\eta\gamma} \xi_i \quad \text{where} \quad \xi_i \sim \mathcal{N}(\mathbf{0}, \mathbf{I}), \tag{F.2}$$

where $\eta$ represents the step size, and $\gamma$ is the noise scale.

**Maximizing the likelihood (ML).** The refinement procedure can alternatively be viewed as a regularized optimization problem aimed at identifying the most probable TS while adhering to specific constraints on the observed time steps. Formally, this is expressed as

$$\arg\min_{\mathbf{y}} \left[-\log p_\theta(\mathbf{y}) + \lambda \mathcal{R}(\mathbf{y}, \tilde{\mathbf{y}})\right], \tag{F.3}$$

which can be optimized using gradient descent.

# G Statistics of Non-stationarity

**Integrated autocorrelation time (IAT) [22].** Integrated autocorrelation time (IAT) quantifies the efficiency of an MCMC sampler by estimating the effective number of independent samples produced by the sampler through calculating the autocorrelation (AC) within a chain. A sampler with a lower IAT is considered more efficient. Additionally, IAT can be used to diagnose the level of noise in a TS, as a TS with higher noise levels will exhibit smaller AC in their absolute values. The IAT and its absolute version (IAAT) are defined as

$$\tau_{\text{IAT}} = 1 + 2 \sum_{k=1}^{\infty} \rho_k, \quad \tau_{\text{IAAT}} = 1 + 2 \sum_{k=1}^{\infty} |\rho_k|, \tag{G.1}$$

respectively, where $\rho_k$ is an AC at lag $k$. Note that this computation needs to be truncated at a practical lag where the ACF values are essentially zero.

**Lag-one autocorrelation (LagAC) [37].** LagAC ($\rho_1$) is a measure of AC that quantifies the correlation between adjacent observations in a TS, which is defined as

$$\rho_1 = \frac{\sum_{t=1}^{n-1}(x_t - \bar{x})(x_{t+1} - \bar{x})}{\sum_{t=1}^{n}(x_t - \bar{x})^2}, \tag{G.2}$$

where $x_t$ and $x_{t+1}$ are consecutive observations in the TS, $\bar{x}$ is the mean of the TS observations, and $n$ is the total number of observations in the TS.

**Variance of autocorrelation (VarAC) [37].** VarAC ($\sigma_\rho^2$) measures the variance of the AC over different lags, which is defined as

$$\sigma_\rho^2 = \frac{1}{m} \sum_{k=1}^{m}(\rho_k - \bar{\rho})^2, \tag{G.3}$$

where $\bar{\rho}$ is the mean of the AC over $m$ lags, and $m$ is the total number of lags considered.

# H Robustness to Statistics of Non-stationarity

Table H.1 presents the CRPS on forecasting tasks with eight datasets using various statistics representing non-stationarity. The result demonstrates our method's robustness across various statistics.

| | | Solar | Electricity | Traffic | Exchange | M4 | UberTLC | KDDCup | Wikipedia | Avg. |
|---|---|---|---|---|---|---|---|---|---|---|
| TSDiff | | 0.399 | 0.049 | 0.105 | 0.012 | 0.036 | 0.172 | 0.335 | 0.221 | 0.166 |
| TSDiff+ANT | VarAC | 0.369 | 0.050 | 0.102 | 0.010 | 0.029 | 0.169 | **0.325** | 0.209 | 0.158 |
| | LagAC | **0.326** | 0.053 | 0.101 | **0.009** | 0.026 | 0.163 | **0.325** | 0.210 | 0.151 |
| | IAAT | **0.326** | **0.047** | 0.101 | **0.009** | 0.026 | 0.163 | **0.325** | 0.206 | **0.150** |
| Oracle | | **0.326** | **0.047** | 0.099 | **0.009** | 0.026 | 0.163 | **0.325** | 0.206 | **0.150** |

Table H.1: Robustness to statistics of non-stationarity.

# I  Robustness to Discrepancy Metric

Tables I.1 and I.2 present the schedules proposed by ANT and the CRPS on forecasting tasks with eight datasets using various discrepancy metrics between the non-stationarity curve and a linear line, respectively. For the metrics, we employ mean-squared error (MSE), mean-absolute error (MAE), Pearson correlation (Corr.), and R-squared ($R^2$), along with the area under the curve (AUC). The result demonstrates our method's robustness across various metrics for d.

| d | Solar | Electricity | Traffic | Exchange | M4 | UberTLC | KDDCup | Wikipedia |
|---|---|---|---|---|---|---|---|---|
| MSE | **Lin(100)** | **Cos(75,2.0)** | Lin(20) | **Cos(50,0.5)** | Sig(100,1.0) | **Lin(20)** | **Lin(50)** | **Cos(75,2.0)** |
| MAE | **Lin(100)** | **Cos(75,2.0)** | Lin(20) | **Cos(50,0.5)** | **Cos(100,1.0)** | **Lin(20)** | **Lin(50)** | **Cos(75,2.0)** |
| Corr. | **Lin(100)** | Sig(100,0.3) | Lin(20) | **Cos(50,0.5)** | Cos(100,0.5) | **Lin(20)** | **Lin(50)** | Cos(100,2.0) |
| $R^2$ | **Lin(100)** | Cos(50,1.0) | Lin(20) | **Cos(50,0.5)** | Sig(100,1.0) | **Lin(20)** | **Lin(50)** | **Cos(75,2.0)** |
| AUC | **Lin(100)** | **Cos(75,2.0)** | Lin(50) | **Cos(50,0.5)** | **Cos(100,1.0)** | **Lin(20)** | **Lin(50)** | **Cos(75,2.0)** |
| Oracle | **Lin(100)** | **Cos(75,2.0)** | **Cos(75,2.0)** | **Cos(50,0.5)** | **Cos(100,1.0)** | **Lin(20)** | **Lin(50)** | **Cos(75,2.0)** |

Table I.1: Robustness of schedule to discrepancy metric.

| | | Solar | Electricity | Traffic | Exchange | M4 | UberTLC | KDDCup | Wikipedia | Avg. |
|---|---|---|---|---|---|---|---|---|---|---|
| TSDiff | | 0.399 | 0.049 | 0.105 | 0.012 | 0.036 | 0.172 | 0.335 | 0.221 | 0.166 |
| | MSE | **0.326** | **0.047** | 0.108 | **0.009** | 0.028 | **0.163** | **0.325** | **0.206** | 0.152 |
| | MAE | **0.326** | **0.047** | 0.108 | **0.009** | 0.026 | **0.163** | **0.325** | **0.206** | 0.151 |
| TSDiff+ANT | Corr. | **0.326** | **0.047** | 0.108 | **0.009** | 0.026 | **0.163** | **0.325** | 0.208 | 0.151 |
| | $R^2$ | **0.326** | 0.049 | 0.108 | **0.009** | 0.028 | **0.163** | **0.325** | **0.206** | 0.152 |
| | AUC | **0.326** | **0.047** | 0.101 | **0.009** | 0.026 | **0.163** | **0.325** | **0.206** | **0.150** |
| Oracle | | **0.326** | 0.049 | **0.099** | **0.009** | 0.026 | **0.163** | **0.325** | **0.206** | **0.150** |

Table I.2: Robustness of CRPS to discrepancy metric.

# J ANT under Constraint on $T$

## J.1 Schedule with Constraint on $T$

ANT can propose a schedule under a constraint on $T$, which offers a flexible trade-off between performance and computational cost. As shown in Table J.1, the proposed noise schedules may vary based on the constraint, highlighting the importance of using an appropriate noise schedule under different resource constraints.

| $T$ | Solar | Electricity | Traffic | Exchange | M4 | UberTLC | KDDCup | Wikipedia |
|---|---|---|---|---|---|---|---|---|
| 10 | Linear(10) | Cos(10,1.0) | Linear(10) | Cos(10,0.5) | Cos(10,0.5) | Linear(10) | Linear(10) | Cos(10,1.0) |
| 20 | Linear(20) | Cos(20,1.0) | Linear(20) | Cos(20,0.5) | Sig(20,1.0) | **Linear(20)** | Linear(20) | Cos(20,2.0) |
| 50 | Linear(50) | Cos(50,1.0) | **Linear(50)** | **Cos(50,0.5)** | Cos(50,1.0) | Linear(50) | **Linear(50)** | Cos(50,2.0) |
| 75 | Linear(75) | **Cos(75,2.0)** | Linear(75) | Cos(75,0.5) | Cos(75,1.0) | Linear(75) | Linear(100) | **Cos(75,2.0)** |
| 100 | **Linear(100)** | Cos(100,2.0) | Linear(100) | Cos(100,0.5) | **Cos(100,1.0)** | Linear(100) | Linear(100) | Cos(100,2.0) |

Table J.1: **Schedule with constraint on $T$.** The schedule proposed by ANT varies based on the constraint on $T$, with the schedules proposed without the constraint in **red**.

## J.2 TS Forecasting with Constraint on $T$

Table J.2 shows the CRPS on forecasting tasks with a constraint of $T^* = 50$ where ANT selects a schedule among candidates with $T$ no larger than $T$ ($T \leq T$). The figure indicates that ANT achieves better performance across all datasets even with no more than half of the steps of the base schedule of $T = 100$.

| TSDiff | | $T$ | Solar | Electricity | Traffic | Exchange | M4 | UberTLC | KDDCup | Wikipedia | Avg. |
|---|---|---|---|---|---|---|---|---|---|---|---|
| - | | 100 | 0.399 | 0.049 | 0.105 | 0.012 | 0.036 | 0.172 | 0.335 | 0.221 | 0.166 |
| +ANT | w/ con. | $\leq 50$ | 0.372 | f.049 | **0.101** | **0.009** | 0.029 | 0.165 | **0.325** | 0.210 | 0.157 |
| | w/o con. | $\leq 100$ | **0.326** | **0.047** | **0.101** | **0.009** | **0.026** | **0.163** | **0.325** | **0.206** | **0.150** |

Table J.2: **Forecasting result with constraint on $T$.** TSDiff applied with ANT under a constraint on diffusion steps ($T \leq 50$) still yields better performance despite the reduced number of steps.

# K Comparison with Other Noise Schedules

Table K.1 shows the comparison of our method with a cosine schedule [24] and a recently proposed method [21] in the computer vision domain that utilizes a schedule enforcing a zero signal-to-noise ratio (SNR) at the terminal step and employs $v$-prediction [27]. Note that we tune the hyperparameters for the above methods using the same range as our method. The results show that our method outperforms the two other methods across all eight datasets in terms of CRPS on forecasting tasks, emphasizing the importance of using an adaptive schedule that considers the rate of data corruption during the forward process.

| Schedule | Solar | Electricity | Traffic | Exchange | M4 | UberTLC | KDDCup | Wikipedia | Avg. |
|---|---|---|---|---|---|---|---|---|---|
| Linear | 0.399 | 0.049 | 0.105 | 0.012 | 0.036 | 0.172 | 0.335 | 0.221 | 0.166 |
| Cosine [24] | 0.346 | 0.048 | 0.102 | 0.010 | **0.026** | 0.174 | 0.368 | 0.209 | 0.160 |
| Zero [21] | 0.441 | 0.055 | 0.123 | 0.010 | 0.046 | 0.184 | 0.387 | 0.240 | 0.185 |
| ANT | **0.326** | **0.047** | **0.101** | **0.009** | **0.026** | **0.163** | **0.325** | **0.206** | **0.150** |

Table K.1: Comparison with other schedules.

## L   IAAT for Multivariate TS

IAAT for multivariate TS can be applied in two different ways: (1) by calculating IAAT for each variable individually, and (2) by calculating a single IAAT that considers all variables together, which we denote as mIAAT (multivariate IAAT).

**Calculation of mIAAT.** First, the autocorrelation function is defined for each pair of variables at different lags. For a multivariate TS $\mathbf{X}_l = (\mathbf{x}_{1l}, \mathbf{x}_{2l}, \ldots, \mathbf{x}_{dl})$ with $d$ variables of length $L$, the $\rho$ at lag $k$ for variable pairs $(i, j)$ is computed as

$$\rho_k^{(i,j)} = \frac{\sum_{t=1}^{L-k} \left(\mathbf{x}_{il} - \bar{\mathbf{x}}_i\right) \left(\mathbf{x}_{j,l+k} - \bar{\mathbf{x}}_j\right)}{\sqrt{\sum_{l=1}^{L} \left(\mathbf{x}_{il} - \bar{\mathbf{x}}_i\right)^2 \sum_{t=1}^{L} \left(\mathbf{x}_{jl} - \bar{\mathbf{x}}_j\right)^2}}, \tag{L.1}$$

where $\bar{\mathbf{x}}_i$ and $\bar{\mathbf{x}}_j$ are the means of the TS for variables $i$ and $j$, respectively.

Then, for each variable $\mathbf{X}^{(d)}$, IAAT is calculated by computing the sum of autocorrelations across all lags and variables:

$$\tau_{\text{IAAT}}^{(i)} = 1 + 2 \sum_{k=1}^{\infty} \sum_{j=1}^{d} \rho_k^{(i,j)}. \tag{L.2}$$

To aggregate the IAAT across multiple variables, a weighted average of the IAAT values for individual variables can be employed. In this method, the weights are determined by the variances of the individual variables, thereby assigning greater importance to variables with higher variance. Consequently, the IAAT for the multivariate TS (mIAAT) can be calculated as follows:

$$\tau_{\text{mIAAT}} = \frac{\sum_{i=1}^{d} \sigma_i^2 \tau_{\text{IAAT}}^{(i)}}{\sum_{i=1}^{d} \sigma_i^2}, \tag{L.3}$$

where $\sigma_i^2$ is the variance of the $i$-th variable.

# M Application to CSDI and Multivariate TS

To demonstrate the effectiveness of our method with other model architectures and multivariate TS, we apply our method to CSDI [34], which is a score-based conditional diffusion model that uses transformer layers for multivariate TS forecasting and imputation. For the experiment, we conduct a forecasting task using Solar [17], M4 [23] and Electricity [17, 3], employing the CRPS as an evaluation metric and a Lin(100) as

| Dataset | $D$ | $L$ | $H$ |
|---|---|---|---|
| Solar | 137 | 168 | 24 |
| Electricity | 370 | 168 | 24 |
| M4 | 414 | 96 | 48 |

Table M.1: Statistics of datasets.

a base schedule. Table M.1 displays the statistics of the datasets, where $D$, $L$, and $H$ represent the number of variables, the length of the input window, and the length of the target window, respectively.

Following the original paper, we normalize each feature to have zero mean and unit variance, and set $\beta_1 = 0.0001$ and $\beta_T = 0.5$ for the schedule. Note that as CSDI adds noise only to certain parts of the TS which are randomly masked with a ratio $r \sim$ Uniform(0,1), we calculate the IAAT with a partially noised TS by randomly masking certain series with ratio $r$ and only performing the forward process on the masked parts, excluding the unmasked parts.

As we use a multivariate TS dataset, we apply our method in two ways: (1) using a common schedule across all variables (with mIAAT) where we compute the ANT score based on Equation L.3, and (2) using a variable-specific schedule for all variables (with IAAT). For the variable-specific schedule, we choose a schedule with a common $T$ for all variables, and Figure M.1 shows the ratio of schedules chosen as adaptive schedules for each variable.

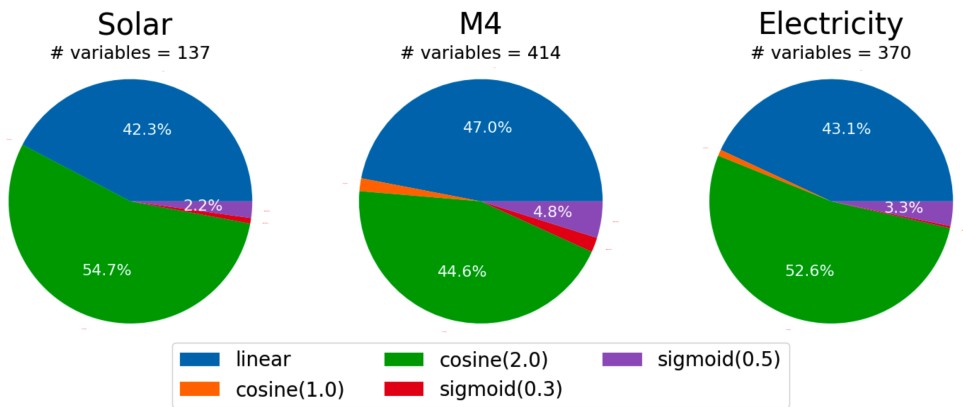

Figure M.1: Ratio of adaptive schedule for each variable.

Table M.2 shows the results of TS forecasting, indicating that our method improves performance when using a common schedule for all variables, but hampers performance when using an adaptive schedule for each variable, underscoring the importance of employing a common schedule for all variables using mIAAT. Furthermore, Figure M.2 shows the non-stationarity curves for all variables of Solar, with the 5th and 95th percentiles shaded, using both the base schedule and the schedule proposed by ANT. It is evident that ANT resembles the ideal linear line (black) more closely.

|  |  | Solar | M4 | Electricity |
|---|---|---|---|---|
| CSDI |  | 0.402 | 0.109 | 0.044 |
| CSDI+ANT | w/ mIAAT | 0.351 | 0.035 | **0.041** |
|  | w/ IAAT | 0.706 | 0.063 | 0.050 |
| Oracle |  | **0.349** | **0.034** | **0.041** |

Table M.2: Multivariate TS forecasting task.

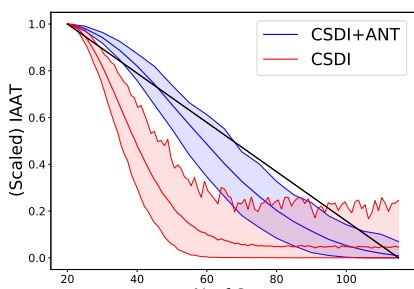

Figure M.2: Non-stationarity curves.

# N Efficiency of ANT

## N.1 Inference Time by $T$

Table N.1 displays the inference time of TSDiff with schedules of different $T$s. Specifically, we report inference time for a TS data averaged on the test dataset of Wikipedia [8]. The results highlight the efficiency of using a smaller $T$, achieving the best forecasting performance (low CRPS) with a schedule of $T = 75$, as proposed by ANT. Notably, the base schedule performs worse despite using a larger $T$ of $T = 100$.

| | Inference | | | | | | |
|---|---|---|---|---|---|---|---|
| | Oracle | | | | | w/ ANT | w/o ANT |
| $T$ | 10 | 20 | 50 | 75 | 100 | 75 | 100 |
| Time (sec) | 0.15 | 0.29 | 0.71 | 1.06 | 1.40 | 1.06 | 1.40 |
| CRPS | 0.225 | 0.210 | 0.207 | **0.206** | 0.208 | **0.206** | 0.221 |

Table N.1: Inference time (sec) by $T$.

## N.2 Comparison of Training & Inference Time

Table N.2 compares the training and inference times of TSDiff with and without the application of ANT across eight datasets, where we train for 1000 epochs and perform inference with the entire test dataset. The results demonstrate that training time is reduced for datasets where linear schedules are selected from ANT, as diffusion step embedding is eliminated. In terms of inference time, efficiency is gained through the reduced $T$. For instance, on the UberTLC dataset, efficiency improves by 78.1% with $T$ reduced from 100 to 20.

| | Train (min) | | | Inference (sec) | | |
|---|---|---|---|---|---|---|
| | w/o ANT | w/ANT | + Gain(%) | w/o ANT | w/ANT | + Gain(%) |
| Solar | 93.7 | 85.2 | **+9.1%** | 1.28 | 1.24 | **+3.4%** |
| Electricity | 95.4 | | +0.0% | 1.28 | 0.96 | **+24.1%** |
| Traffic | 100.5 | 92.0 | **+8.5%** | 1.28 | 0.64 | **+50.0%** |
| Exchange | 92.0 | | +0.0% | 1.48 | 0.68 | **+46.2%** |
| M4 | 85.2 | | +0.0% | 1.20 | | +0.0% |
| UberTLC | 94.5 | 86.0 | **+9.0%** | 1.28 | 0.27 | **+78.1%** |
| KDDCup | 106.5 | 99.6 | **+6.5%** | 1.24 | 0.62 | **+49.4%** |
| Wikipedia | 98.9 | | +0.0% | 1.40 | 1.06 | **+24.0%** |

Table N.2: Train & Inference time w/ and w/o ANT.

# O    TS Generation with Electricity

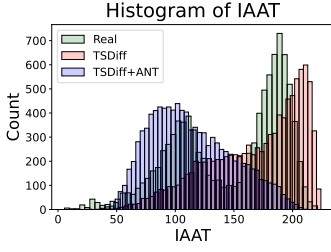
(a) Histogram.

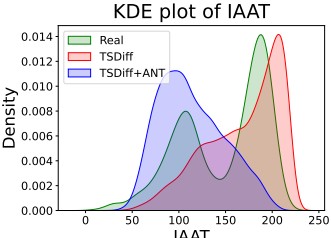
(b) Kernel density estimation.

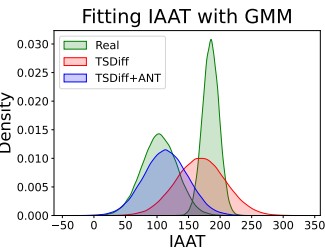
(c) Gaussian mixture model.

Figure O.1: IAAT plots of real and generated TS.

| Data | | | Modality | | Parameters | Ratio |
|------|--|--|----------|--|------------|-------|
| Real | | | Multi | 1) | $N(103.8, 28.1^2)$ | 45.6% |
|      | | |       | 2) | $N(185.3, 13.1^2)$ | 54.4% |
| TSDiff | w/o ANT | | Uni | | $N(169.0, 39.5^2)$ | 100% |
|        | w/ ANT  | |     | | $N(113.8, 34.9^2)$ | 100% |

Table O.1: Result of Gaussian fitting for IAATs.

| Data | | CRPS | | | JSD |
|------|--|------|--|--|-----|
|      | | Linear | DeepAR | Trans. | |
| Real | | 0.088 | **0.054** | 0.076 | - |
| TSDiff | w/o ANT | **0.065** | 0.058 | **0.056** | 0.343 |
|        | w/ ANT  | 0.067 | 0.058 | 0.061 | 0.369 |

Table O.2: CRPS & JSD with Real.

Figure O.1a and O.1b display the IAAT distributions of real TS and TS samples generated from TSDiff using Electricity [17, 3], with and without ANT, using histogram and kernel density estimation (KDE) plots, respectively. The figures indicate that while the distribution of real TS exhibits multimodal characteristics, TSDiff without ANT captures only one modality, whereas TSDiff with ANT captures the other modality. Figure O.2 presents a visualization of real TS by the IAAT, exhibiting distinct patterns among different TS. TS (A), characterized by the lowest IAAT, exhibits no discernible patterns, whereas TS (E), with the highest IAAT, demonstrates a pronounced seasonality.

To assess the similarity between the real TS and the generated TS, we employ Gaussian mixture model (GMM) fitting on the IAAT distributions, with the results presented in Figure O.1c. Table O.1 provides the mean and variance of each modality, revealing that the modality captured by TSDiff without ANT accounts for 45.6% of the data, while the modality captured by TSDiff with ANT constitutes 54.4% of the dataset.

To analyze the the performance of TS generation task in terms of the IAAT distribution, we calculate the Jensen–Shannon divergence (JSD) between the IAAT distributions of the real TS and the generated TS. Taking into account the ratio of data in each modality, we compute the weighted JSD with weights derived from Table O.1. The results in Table O.2 indicate that TS generated from TSDiff without ANT exhibits a lower JSD compared to those generated with ANT, resulting in slightly lower average CRPS across three different base forecasters (Linear, DeepAR, and Transformer).

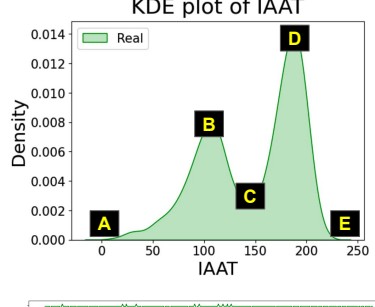

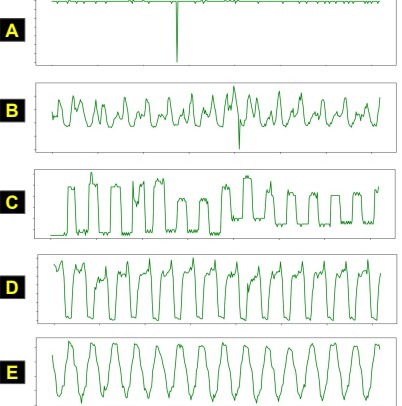

Figure O.2: Visualization by IAAT.

It's noteworthy that the performance is the worst with the real samples, suggesting the inability of the forecasters to predict TS with different modalities using a single model. This also suggests that the quality of the synthetic dataset cannot be fully captured by the TS generation task of train-synthetic-test-real (TSTR), indicating its insufficiency.

# P t-SNE Visualizations

Figure P.1 depicts the t-SNE visualizations of embeddings of CNN features extracted from a 1D-CNN encoder of the proxy classification model, with both linear and non-linear schedules. The figure shows that both visualizations of the linear and non-linear schedules yield directional point movement across all output channels as the step progresses. However, with the linear schedule, the directions appear more diverse, indicating that step information is captured by both schedules, but more limited with a non-linear schedule.

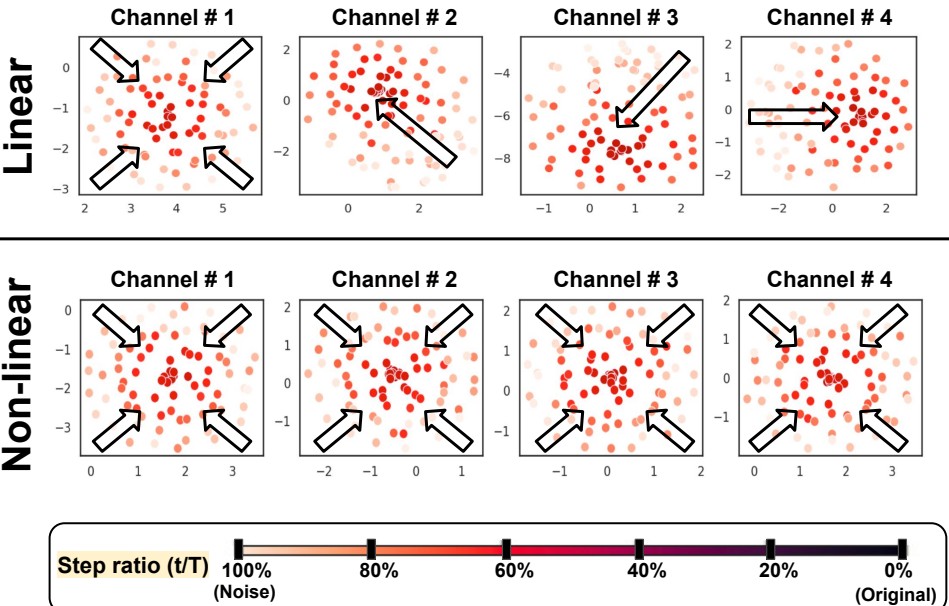

Figure P.1: t-SNE visualizations with linear and non-linear schedules.

# Q  Visualization of Non-stationarity Curves

Figure Q.1 presents the visualization of the non-stationarity curves of eight datasets when employing a base schedule of TSDiff (Lin(100)), and adaptive schedules proposed by ANT. The result indicates that, in general, the larger the reduction in discrepancy between the linear line and the non-stationarity curve by the proposed schedule, the greater the performance gain achieved. Note that for the Solar [17] dataset, the schedule proposed by ANT is identical to the base schedule, and any performance gain is solely attributed to the elimination of the diffusion step embedding.

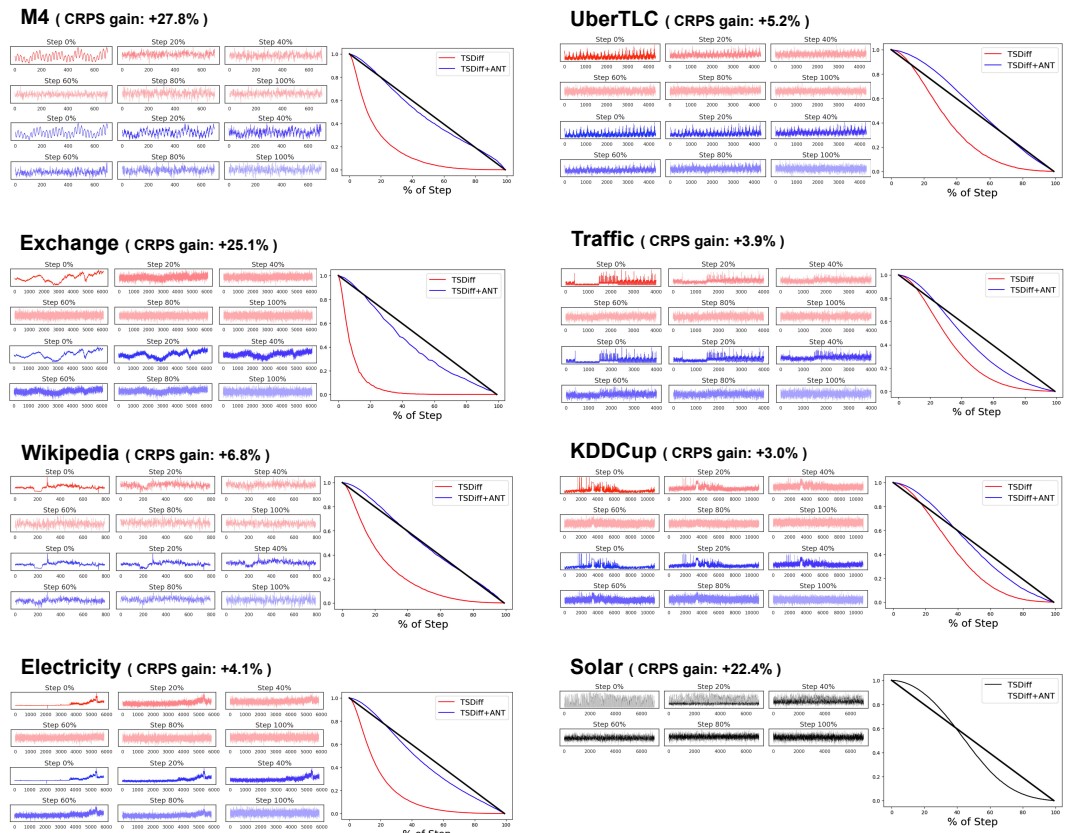

Figure Q.1: **Visualization of non-stationarity curves.** The figure shows the non-stationarity curves of eight datasets, when employing a base schedule and and an adaptive schedule proposed by ANT.

