# OpenReview forum: "ANT: Adaptive Noise Schedule for Time Series Diffusion Models"
_NeurIPS.cc/2024/Conference — NeurIPS 2024 poster_

### Official Review · Reviewer_xQJK · 2024-06-30

**Soundness:** 3
**Presentation:** 2
**Contribution:** 3
**Rating:** 7
**Confidence:** 4

**Summary:**

Paper introduces the new method, adaptive noise schedule  for time series diffusion model(ANT) . This is a new methodology as per my knowledge and hence extends the boundaries of the field. Diffusion models are highly effective in generating data but their inclusion in Time Series tasks overlooks the significance of properties of time series and don't focus on dedicated noise scheduling.

The paper introduces an algorithm to choose an adaptive noise scheduling which has not been done in the past for time series. Authors clearly define the statistics used to quantify the non-stationarity and ANT score. The proposed schedule noise aims to corrupt the TS data gradually instead of abruptly as done with Linear scheduling. It also demonstrates the need for different scheduling needed for different datasets. One size fit all solution does not serve the purpose.

Another important aspect of the paper discusses and proves why diffusion embedding is not needed for linear schedule. In addition, there was several analysis which shows the importance and stability of ANT.

**Strengths:**

Originality: The paper introduce new method ANT for adaptive noise scheduling for time series. This is an original idea in TS domain. In addition the a new metric is introduced (IIAT) to quantify the importance of ANT.

Quality: Overall the quality of the paper is good, all the statements/claims are backed with experimental results. Authors conducted extensive experiments and analysis across several datasets and tasks which However, the details about the experiments/analyses and evaluation is limited and can be explained further in details.

Clarity: The paper is well structured, clearly written making is accessible for reading to different level of background knowledge with diffusion model for TS. The new algorithm, metric and concepts are introduced clearly in the draft under review. The terms are defined and experimented are backed with tables and figures which make it clear to understand the new algorithm better.

Significance: ANT significantly improve the accuracy of the TS forecasting models by used adaptive noise scheduling and have wide applications. The paper describes how non-stationarity can be used to enhance the performane for generative tasks.

**Weaknesses:**

Although paper is well structures and provide tables and figured for all the experiments conducted, details on the evaluation of experiments would help the reader further. E.g. How is metric calculated for the time series forecasting task; what fraction of data is used for inference and which evaluation method is used.

**Questions:**

1. There are several places in the text where language can be improved to avoid ambiguity and enhance the clarity of paper.  E.g. L27 text: "Several works have been proposed to design appropriate schedules [24, 21], but they are not tailored to the TS domain, considering the characteristics of TS data.". L31: it should be M4 dataset and a reference added to it.

2. Conclusion/Summary does not discuss/report the main results quantitatively.

**Limitations:**

Authors does mention about the limitation at the end of the draft that ANT could take long time if to find the noise schedule if there are too many candidate schedules and this motivated future work in this direction.   It is also noted that the comparison for forecasting task is done using very limited models to show the effectiveness of the ANT but more advanced models (DLinear/FEDFormer etc) could be used in comparison for completeness and also give a hint how big is the difference in metric.

---

> ### Author Rebuttal · Authors · 2024-08-06
>
> ### **W1. Details on the evaluation of experiments**
> We agree that some details on the evaluation of experiments were omitted due to space limitations. Here, we provide the additional details and will incorporate them into Appendix.
>
>
> &nbsp;
>
>
> > 1. **Metric for TS forecasting task**
>
>
>
>
> Following prior works on TS diffusion models [6,18,19,26,34,38] including TSDiff [15], we utilize the **Continuous Ranked Probability Score (CRPS)** [9] to assess the quality of *probabilistic* forecasts, where we describe the details in **Appendix B**. Note that TS diffusion models predict the *probabilistic* distribution rather than the *deterministic* value, so CRPS accounts for the uncertainty of the predicted results, unlike MSE or MAE.
>
>
> &nbsp;
>
>
> > 2. **Fraction of (test) data used for inference**
>
>
>
>
> All datasets used in our experiments are sourced from GluonTS [2], where training and test splits are provided. We follow the setting of TSDiff [15], where the validation set is created by splitting a portion of the training dataset, with the split ratio determined by the size of the train and test datasets.
> Further details about the training and test sets are provided in **Table A.1**.
>
>
> &nbsp;
>
>
> > 3. **Evaluation method**
>
>
>
>
> For evaluation, we calculate the CRPS using only the **test dataset**. Specifically, we follow the approach used in TSDiff [15] by utilizing the `gluonts.transform.sampler.TestSplitSampler` from GluonTS [2], which leverages the final time steps of the test TS instances to predict future values.
>
>
> &nbsp;
>
>
> ### **Q1. Clarifying sentences and adding references**
> Thank you for addressing this. We are planning to improve overall writing in the revision for better clarity including your suggestions. For sure, what we meant in **L27** is that noise schedules used in [24, 21] are not tailored to the TS domain (but images). Regarding M4 in **L31**, we have already mentioned **the M4 dataset** with a reference in the previous sentence (**L28**), but we agree that the term **M4** might be confusing when used standalone.
>
>
> &nbsp;
>
>
> ### **Q2. Main quantitative results missing in the conclusion.**
> Thank you for pointing this out. As we have emphasized our results quantitatively in the introduction, we will also summarize and reiterate the main findings quantitatively in the conclusion section.
>
>
> &nbsp;
>
>
> ### **L1-2. Comparison with more advanced models (DLinear/FEDFormer etc).**
>
>
> > **1) Application to CSDI**
>
>
>
>
> We appreciate the reviewer xQJK's concern about the **limited range of models** used to demonstrate the effectiveness of ANT. We completely agree and, as discussed in **Appendix M**, we have *already* applied our method to another well-known TS diffusion model, CSDI [34]. We believe this further validates the applicability and effectiveness of our method across different TS diffusion models (in both univariate and multivariate settings).
>
>
> &nbsp;
>
>
> > **2) Comparing with other TS forecasting models [A,B]**
>
> In short, DLinear [A] and Fedformer [B] are not compared because they are for *deterministic* forecasting, while our task is *probabilistic* forecasting; TS forecasting models for these two different tasks have been developed independently.
> We provide more detailed reasons as below:
> - **Model**) TS diffusion models, including TSDiff, are probabilistic forecasting models, while models like DLinear and Fedformer are deterministic forecasting models, which have developed independently in prior works.
> - **Metric**) For the above reason, previous works on TS diffusions [6,18,19,26,34,38] do not use them as baselines and instead employ a different metric, CRPS, to account for the uncertainty of the predicted results, rather than the MSE/MAE metrics commonly used in deterministic models.
> - **Dataset**) TS diffusion models are mostly designed for short-term TS forecasting (STSF) [C], whereas models like DLinear and Fedformer are intended for long-term TS forecasting (LTSF). In LTSF tasks, datasets generally consist of long TS with fewer instances, and training and testing splits are usually based on different time periods of the same instances. In contrast, STSF tasks involve shorter time series with more instances, and the split is based on different instances of the TS.
>
>
>
>
> Nonetheless, to address the reviewer xQJK's concern, we conducted an additional experiment with DLinear [A] using the Solar dataset [17]. Since DLinear is a deterministic model and cannot be directly compared with our task which uses CRPS as a metric, we employ a common technique for probabilistic forecasting [28] by predicting the mean and standard deviation of a normal distribution rather than exact values. The results show that DLinear yields a CRPS of **0.660**, while TSDiff and {TSDiff with ANT} achieve CRPS scores of **0.399** and **0.326**, respectively.
>
> &nbsp;
>
> [A] Zeng, Ailing, et al. "Are transformers effective for time series forecasting?" AAAI 2023
>
>
> [B] Zhou, Tian, et al. "Fedformer: Frequency enhanced decomposed transformer for long-term series forecasting." ICML 2022
>
>
> [C] Meijer, Caspar, and Lydia Y. Chen. "The Rise of Diffusion Models in Time-Series Forecasting." arXiv 2024

---

### Official Review · Reviewer_eJYD · 2024-07-12

**Soundness:** 2
**Presentation:** 3
**Contribution:** 3
**Rating:** 6
**Confidence:** 2

**Summary:**

This paper proposes ANT, which adaptively selects the optimal noise schedule for time series diffusion models. The ANT score is computed for each schedule offline based on the datasets' statistics, which is the basis for selection. Extensive experiments demonstrate the method's superior performance in multiple time series tasks.

**Strengths:**

- This method is innovative. Different from previous work that directly applied the existing general framework, it takes into account the characteristics of time series data, that is, non-stationarity, to select the scheduler.
- The experiments cover a variety of tasks and data sets, and the experimental results are convincing.

**Weaknesses:**

- These candidate schedules are manually set in advance and are not guaranteed to include the optimal schedules.
- The ANT score is a simple multiplication of three factors, which is relatively intuitive. Their high or low values ​​cannot guarantee the quality of the schedules.

**Questions:**

- What is the basis for the design of your candidate schedule? What if they do not include the optimal schedule? Will it have a big impact on the results of the experiment?
- Although a lower score indicates a better schedule, are there many cases where schedules with higher scores have better results?
- Are the optimal schedules for the same data set significantly different between different diffusion methods? Can the proposed method consider the characteristics of specific methods when selecting a schedule?

**Limitations:**

Yes

---

> ### Author Rebuttal · Authors · 2024-08-06
>
> ### **W1, Q1. Candidate schedules do not guarantee to include the optimal schedule.**
> As the reviewer eJYD mentioned, the candidate schedule may not include the optimal schedule. However, we note that we do **not aim to find the optimal schedule**; our proposed ANT is a **criterion for choosing a better noise schedule from candidates based on the characteristics of the dataset**.
>
> For experiments, we selected three widely used schedules (linear, cosine, sigmoid) [4] as our candidates, incorporating five different diffusion steps and three different temperatures for non-linear schedules. This resulted in a total of **35 schedules**, which we believe are sufficient, as they encompass most of the schedules used in previous TS diffusion studies [1,26,29,30].
>
> As shown in **Figure 11** and **Table 12**, ANT is applicable to non-trivial schedules, i.e., it gives a higher score to sensibly designed noise schedules if they perform better than trivial ones. However, as mentioned in the conclusion, finding the optimal schedule for a given dataset using our ANT score (via optimization) remains a future work.
>
> &nbsp;
>
> ### **W2. Relationship between ANT score and the quality of schedule**
> The reviewer raised concerns about the relationship between the ANT score and the quality of the schedule. However, the proposed ANT score aligns with the three desiderata of noise schedules, which can be considered indicators of schedule quality. These desiderata have been discussed in prior works and our paper:
>
> - **1) $\lambda_{\text{linear}}$: Reducing non-stationarity on a linear scale**
>
> Previous work of DDPM [24] proposed a cosine schedule (instead of linear) and argued that maintaining a consistent noise level at each step leads to better quality, as mentioned in **L38--39**.
>
> - **2) $\lambda_{\text{noise}}$: Corruption to random noise**
>
> Previous work [21] suggested that the schedule must be capable of corrupting the TS into random noise at the final step to ensure the generation of high-quality samples, as the reverse process (sampling) of the diffusion model begins with random noise, as mentioned in **L40--41** and **L133--135**.
>
> - **3) $\lambda_{\text{step}}$: Sufficient steps**
>
> Previous work of consistency models [31] emphasized the need for a sufficient number of steps to generate high-quality samples, as mentioned in **L41--42** and **L135--136**.
> Furthermore, as shown in **Table 7**, using all the three components of ANT mostly results in the oracle, backing up our argument. For example, with the M4 dataset [23], using all three components results in a value of 0.026 with a Cosine(100,1.0) schedule, whereas the ANT score without considering $\lambda_\text{noise}$ yields a CRPS of 0.094 with a Cosine(10,0.5) schedule. Notably, the base schedule of TSDiff (Linear(100)) yields a CRPS of 0.036.
>
> &nbsp;
>
> ### **Q2. Case for higher ANT score but with better result.**
> A lower ANT score generally indicates a better schedule, as shown in **Figure 8(a)**, and this tendency becomes stronger when using all three components of the ANT score. However, as the reviewer has noted, there can be exceptions where better performance is achieved with a higher ANT score, maybe because of the stochasticity of the forward diffusion process when computing ANT. Nonetheless, the schedule with the lowest ANT score should return a reasonably good performance.
> In our extensive analysis, we found only one such case with the Traffic dataset [3], as shown in **Table 5**. Among the 35 candidate schedules, Linear(50) had the lowest ANT score, yet the best performance was obtained with Cosine(75, 2.0), with a marginal difference in CRPS(0.101 vs. 0.099). It is worth noting that the base schedule Linear(100) yielded a CRPS of 0.105, which is noticeably higher than both Linear(50) and Cosine(75, 2.0).
>
> &nbsp;
>
> ### **Q3. Optimal noise schedule depending on diffusion models**
> Note that the ANT score is computed **without access to the diffusion model**, such that the schedule is selected based on **the characteristics of the dataset**, rather than the **specific design choice on the model architecture** being used.
>
> Nonetheless, as the reviewer eJYD concerned, some specific design choices for diffusion models could affect the optimality of the schedule. As we had a similar concern, in **Appendix M**, we have applied our method to another well-known TS diffusion model, CSDI [34], where we found that ANT is effective for this model as well. We believe this further validates the applicability and effectiveness of our method across different TS diffusion models.

---

> > ### Comment · Reviewer_eJYD · 2024-08-08
> >
> > Thank you for answering my questions in detail. My concerns have been addressed, so I decide to improve my rating to 6.

---

> > > ### Author Response · Authors · 2024-08-08
> > >
> > > Thank you for raising your score!
> > >
> > > If you have any further questions or suggestions, please feel free to share them with us.

---

### Official Review · Reviewer_YUJG · 2024-07-23

**Soundness:** 3
**Presentation:** 3
**Contribution:** 2
**Rating:** 6
**Confidence:** 4

**Summary:**

This paper addresses the non-stationarity in time series data by proposing an ANT score to enable an adaptive noise schedule for diffusion models. It provides extensive experimental results on time series forecasting and generation tasks.

**Strengths:**

1. The idea of adaptively select noise schedule in diffusion models is original and interesting, which can effectively addresses the number of steps issue.
2. The paper introduces an ANT score to enable an adaptive noise schedule, which measures the discrepancy between a linear line and the non-stationary curve.
3. Experiments have been conducted on several different time series tasks including time series forecasting, refinement, and generation, which is generally solid.

**Weaknesses:**

1. Though the ANT score seems effective in empirical evaluations, the paper does not provide theoretical analysis on how the score plays a role in addressing the proposed limitations of traditional diffusion models.
2. The proposed ANT noise schedule can be applied to different types of diffusion models, however in the experiments the ANT is only adapted to TSDiff, which lacks enough evidence to demonstrate its generalizability.

**Questions:**

1. I wonder whether the authors can provide more rigorous theoretical analysis on ANT score.
2. It would be nice to see whether the ANT can be adapted to other diffusion models.

**Limitations:**

The paper limits the noise functions to linear, cosine, and sigmoid, and it may be more imperative to be extended to other cases.

---

> ### Author Rebuttal · Authors · 2024-08-06
>
> ### **W1, Q1. Theoretical analysis on the ANT score**
>
> Previous works on noise schedules have demonstrated their effectiveness primarily through **empirical justification** rather than theoretical analysis [4,21,24]. In line with this approach, we have conducted extensive experiments to support our proposal, as highlighted by reviewers xQJK and eJYD.
>
> Furthermore, following the previous works regarding the schedules, we focus on the intuitive aspects that schedules should meet, where the proposed ANT score aligns with the three desiderata of noise schedules:
> - **1) $\lambda_{\text{linear}}$: Reducing non-stationarity on a linear scale**
>
> Previous work of DDPM [24] proposed a cosine schedule (instead of linear) and argued that maintaining a consistent noise level at each step leads to better quality, as mentioned in **L38--39**.
>
> - **2) $\lambda_{\text{noise}}$: Corruption to random noise**
>
> Previous work [21] suggested that the schedule must be capable of corrupting the TS into random noise at the final step to ensure the generation of high-quality samples, as the reverse process (sampling) of the diffusion model begins with random noise, as mentioned in **L40--41** and **L133--135**.
>
> - **3) $\lambda_{\text{step}}$: Sufficient steps**
>
> Previous work of consistency models [31] emphasized the need for a sufficient number of steps to generate high-quality samples, as mentioned in **L41--42** and **L135--136**.
>
> While the advances in noise scheduling for diffusion models has mostly been made empirically, we hope our work motivates future works on the theoretical understanding of a relationship between optimal noise scheduling and the characteristics of the datasets.
>
> &nbsp;
>
> ### **W2, Q2. Application to other diffusion models.**
>
> We appreciate the reviewer YUJG's concern about the **limited range of models** used to demonstrate the effectiveness of ANT. We completely agree and, as discussed in **Appendix M**, we have *already* applied our method to another well-known TS diffusion model, CSDI [34]. We believe this further validates the applicability and effectiveness of our method across different TS diffusion models (in both univariate and multivariate settings).
>
> &nbsp;
>
> ### **L1. Functions used for candidate schedules.**
> We acknowledge the reviewer YUJG's point on expanding the range of noise functions. However, our focus is **not on identifying the optimal schedule across various noise functions**, but rather on proposing the criterion of choosing effective noise schedules tailored to the dataset's characteristics.
>
> However, we emphasize that the linear, cosine, and sigmoid functions are well-established and widely used in literature [4], and recent TS diffusion models [1, 26, 29, 30] primarily employ linear or cosine schedules. Although developing the best noise function is not our primary goal, we have explored extending our approach with a more flexible noise (and non-trivial) function, such as an ensemble of cosine functions, as demonstrated in **Figure 11** and **Table 12**, to illustrate the potential applicability of ANT for future advancements in noise functions.

---

> > ### Comment · Reviewer_YUJG · 2024-08-09
> >
> > Thanks for addressing my concerns and I decide to raise the score.

---

> > > ### Author Response · Authors · 2024-08-09
> > >
> > > Thank you for raising your score!
> > >
> > > If you have any further questions or suggestions, please feel free to share them with us.

---

### Author Rebuttal · Authors · 2024-08-06

# General Comments
First of all, we deeply appreciate your time and effort in reviewing our paper.
Our work introduces **ANT** (**A**daptive **N**oise Schedule for **T**ime Series), a method for automatically predetermining proper noise schedules based on the characteristics of the dataset.


As agreed by all reviewers, our proposed ANT is original/novel and its effectiveness is supported by extensive experiments. In our responses, we addressed the concerns raised by the reviewers and supplemented our claims with additional analyses. Here are some key highlights to assist with your post-rebuttal discussion:


&nbsp;


### **1. Application to other TS diffusion models (All)**

We found that reviewers are interested in whether ANT is applicable to other TS diffusion models, and we have *already* applied our method to another well-known TS diffusion model, CSDI [34], as detailed in **Appendix M** and guided in **L83--84**.


&nbsp;



### **2. Comparison with other baselines (Reviewer xQJK)**

Reviewer xQJK asked us to compare other baseline methods like DLinear and FEDformer with ours. However, they are proposed for *deterministic* forecasting, while our task with diffusion models is *probabilistic* forecasting, such that they have been developed independently and are not directly comparable. Nonetheless, to address reviewer xQJK's concern, we conducted an additional experiment with DLinear on the Solar dataset and confirmed the superiority of ours (with TS diffusion models) over DLinear.


&nbsp;


### **3. Finding the optimal schedule from finite number of candidates (Reviewer eJYD, Reviewer YUJG)**

We emphasize that our contribution is *not on finding the optimal schedule*, but on proposing a ***criterion for efficiently selecting a better noise schedule from candidates based on the dataset's characteristics***, where we used candidate functions that are well-established and widely used in literature [4]. We also confirmed that our ANT score is able to assign a high score to non-trivial schedules if they result in good performance in **Figure 11** and **Table 12**. We leave the way to find the optimal schedule for a given dataset using our ANT score (via optimization) as a future work.

&nbsp;


### **4. Theoretical analysis on the ANT score (Reviewer YUJG)**

We note that previous works on noise schedules primarily relied on empirical justification rather than theoretical analysis [4, 21, 24], and we have also supported our proposal through extensive experiments following these works. While the advances in noise scheduling for diffusion models has mostly been made empirically, we hope our work inspires future research into the theoretical understanding of the relationship between optimal noise schedule and dataset characteristics.

&nbsp;


We sincerely appreciate the reviewers' valuable feedback and insights. We believe these will significantly enhance the contribution of our paper, and we will integrate them into the final version. Should there be any additional points we may have overlooked or if you have further questions or suggestions, please let us know; we are eager to address them and refine our work.

&nbsp;

Thank you very much.

Authors.

---

### Decision · Program_Chairs · 2024-09-25

**Decision:**

Accept (poster)

**Comment:**

This paper introduces a novel methodology to build time series diffusion models. All the reviewers are positive about the contribution. The authors should incorporate the reviewers' feedback in the final version of their paper.